

# Multi-Source Remote Sensing-Based Reconstruction of Glacier Mass Changes in Southeastern Tibet Since the 21st Century

Zihao Li[1], Qiuyu Wang[1]*, Huan Xu[1], Wei Yang[2], Wenke Sun[1]

[1]State Key Laboratory of Earth System Numerical Modeling and Application, College of Earth and Planetary Sciences, University of Chinese Academy of Sciences, Chinese Academy of Sciences, Beijing, China,
[2]State Key Laboratory of Tibetan Plateau Earth System, Environment and Resources and (TPESER), Institute of Tibetan Plateau Research, Chinese Academy of Sciences, Beijing, China

*Correspondence to*: Qiuyu Wang (wangqiuyu@ucas.ac.cn)

**Abstract.** Glaciers are a crucial freshwater resource and a key indicator of climate change. However, tracking annual changes in glacier inventories remains a significant challenge due to persistent cloud cover and seasonal snow accumulation. The increasing availability of satellite data, particularly from the Sentinel series, has greatly enhanced glacier monitoring capabilities. In this study, we developed an ensemble learning-based random forest classifier using data from Landsat, Sentinel-1, Sentinel-2, and NASADEM to automatically delineate glacier extents in southeastern Tibet from 2016 to 2022, achieving the first annual-resolution glacier inventory in the region. To extend the time series to 2000, we manually constructed glacier inventories for 2000, 2005, 2010, and 2015 by integrating a three-year dataset centered on each target year, addressing the limitations posed by the absence of early Sentinel data. Our results reveal a consistent decline in glacier area, from $7898.61 \pm 652.15$ km$^2$ in 2000 to $6317.13 \pm 592.57$ km$^2$ in 2022, with an average annual loss of $85.03 \pm 7.60$ km$^2$/y. Notably, the retreat rate accelerated after 2010, increasing from $57.72 \pm 16.81$ km²/y (2000–2010) to $97.72 \pm 17.67$ km²/y (2010–2022). By integrating satellite altimetry data, we calculated the glacier mass balance using dynamically updated glacier areas, resulting in an annual mass loss $6.20 \pm 1.16$ Gt/y. Correlation analysis between glacier thickness and area changes showed a strong positive relationship ($R^2 = 0.89$, $p < 0.001$). This study provides a novel approach to high-temporal-resolution glacier assessments by incorporating annual dynamic glacier areas into mass balance calculations. The improved accuracy of these estimations offers a refined understanding of cryosphere changes in southeastern Tibet, underscoring the urgency of monitoring glacier dynamics in response to climate change.

## 1 Introduction

As a vital component of the cryosphere (Cogley et al., 2010), glaciers are among the best natural indicators of climate change and play a crucial role in regulating global sea levels and water resources (Zemp et al., 2019). Glacier mass loss accounts for approximately 20% of the total observed sea-level rise (Zemp et al., 2025), while the Greenland Ice Sheet contributes about 17% (Otosaka et al., 2023). The Tibetan Plateau, known as the "Roof of the World" and the "Water Tower of Asia" (Immerzeel and Bierkens, 2012; Lu et al., 2005), contains the highest concentration of glaciers outside the polar regions (Yao, 2008). It is the source of many major rivers in China and a critical contributor to surface runoff (Lutz et al.,



2014; Smith and Bookhagen, 2018; Pritchard, 2019). Global changes significantly affect regional climate, rivers and lakes evolution, and the formation of geological hazards (Immerzeel et al., 2013; Richardson and Reynolds, 2000; Kääb et al., 2018). The southeastern Tibetan Plateau, in particular, exhibits the most pronounced glacier retreat compared to other areas of the plateau (Yao et al., 2012; Ye et al., 2017). Strongly influenced by the Indian monsoon, these glaciers exhibits high accumulation and ablation rates, making it one of the main regions for maritime glaciers in China (Su and Shi, 2002). These glacier changes are highly sensitive to climate variability, underscoring their role as critical indicators of climatic shifts (Yang et al., 2010b; Sakai and Fujita, 2017)

Satellites imagery-based glacier area mapping can be achieved by two main approaches: visual interpretation and computer-based automated classification. Visual interpretation offers high accuracy and ease of use but requires substantial manpower and resources (Alifu et al., 2015; Hall et al., 1992). Computer-based automated classification methods for remote sensing data include the Normalized Difference Snow Index (NDSI), band ratio methods (Liu et al., 2015), and supervised and unsupervised classifications (Pope and Rees, 2014). Among these, NDSI is a widely used method for extracting glacier and snow cover. It utilizes the ratio of the difference to the sum of the green band and shortwave infrared band, leveraging the high reflectance of glacier in the visible spectrum and their low reflectance in the near-infrared spectrum (Hall et al., 1987; Raup et al., 2007). Huang Lei et al. (Huang et al., 2021) proposed a "multi-temporal minimum NDSI composite" method based on Landsat remote sensing images, using the Google Earth Engine cloud platform to extract bare ice areas in the High Mountain Asia (HMA) region from 1990 to 2018, and analysing the spatiotemporal changes in glacier coverage. Similarly, Keshri et al. (2009) (Keshri et al., 2009) systematically identified debris-covered glaciers using ASTER data by integrating three normalized indices. their method demonstrated the effectiveness of combining these indices in distinguishing snow, ice, and debris-mixed areas. While the normalized index method is highly efficient and accurate for glacier extraction, traditional threshold selection is manual and lacks automation.

The recent surge in machine learning algorithms has expanded their applications in glacier remote sensing. Y. Lu et al. (Lu et al., 2021) proposed an composite classification model for glacier regions by combining the advantages of random forest and convolutional neural networks (CNN). Alifu et al. (Alifu et al., 2020) conducted a comparative study of six classifiers, including K-nearest neighbors (KNN), support vector machines (SVM), gradient-boosting decision trees (GBDT), decision trees (DT), random forests (RF), and multilayer perceptron (MLP). They found that the RF classifier achieved the highest classification accuracy (97%) in effectively retrieving debris-covered glacier information. Xie Fuming et al. (Xie et al., 2020)used Landsat TM and OLI optical images on the Google Earth Engine platform, employing Otsu thresholding combined with machine learning algorithms to extract debris-covered glaciers in the Hunza Basin, Karakoram Mountains. Their approach achieved a Kappa coefficient of $0.94 \pm 0.01$ and an overall classification accuracy of $95.5 \pm 0.9\%$. These studies highlight the promising applications of machine learning in glacier mapping. Ensemble learning methods, particularly random forest, have consistently demonstrated superior performance.



Cloud cover and snow presence are two main challenges for the automatic extraction of glacier information from satellite
imagery. Summer satellite images are generally preferred for identifying glaciers, as most seasonal snow has melted by this
time. However, some glacier in the Tibet areas experience frequent summer snowfall and persistent cloud cover,
complicating this process. Although clouds can be identified in single images through the combination of multispectral
information, visual interpretation alone makes it difficult to merge cloud-free images pixel by pixel. At the same time, the
glacial ablation zone in the region is covered by a large number of surface moraines (Yang et al., 2010a), and the spectral
information of surface moraines and bare soil are similar and easily misclassified. Consequently, generating high temporal
resolution glacier inventory in southeastern Tibet remains a significant challenge.

The most recent comprehensive glacier inventory for southeastern Tibet was conducted by Ye et al (Qinghua, 2020).. they
used 210 scenes of Landsat 8 OLI multispectral remote sensing data (2013–2018), combined with SRTM DEM V4.1 data,
Google Earth imagery, and HJ1A/1B satellite images from different seasons of the same year. This work also incorporated
data from China's first and second glacier inventories to produce the 2017 Tibetan Plateau Glacier Inventory (TPG2017).
Additionally, the team also created glacier inventories for the Tibetan Plateau in 1976, 2001, and 2013 using the same
method (Qinghua, 2019; Wu Yuwei, 2018; Ye et al., 2017).

Other Glacier inventories data relevant to southeastern Tibet include the Southeastern Qinghai–Tibet Plateau Glacier
Inventory (SEQTPGI), created by Ke et al. (Ke et al., 2016) using Landsat and ALOS SAR images (2011–2013), as well as
SRTM DEM data. Broader datasets include the World Glacier Inventory (WGI), Randolph Glacier Inventory (RGI)
(Consortium, 2023), Hindu Kush Himalayan (HKH) (Bajracharya and Shrestha, 2011; Bolch et al., 2012), Glacier Area
Mapping for Discharge from the Asian Mountains (GAMDAM) (Nuimura et al., 2015), MODIS Persistent Ice (MODICE)
(Painter et al., 2012), and the first and second Chinese Glacier Inventories (CGI1 and CGI2) (Shi et al., 2009; Guo et al.,
2015).

Despite these efforts, recent changes in glacier area remain poorly understood. Most datasets represent single-period
observations with varying acquisition seasons and processing methods, complicating reliable comparisons across studies,
epochs, and regions for change analysis. Automated annual-resolution glacier mapping has not yet been achieved. However,
the rapid development of open-access datasets on the Google Earth Engine (GEE) (Gorelick et al., 2017) platform and the
increasing availability of free or low-cost cloud computing infrastructure offer promising opportunities for creating high-
temporal-resolution glacier inventories.

In the past two decades, three primary satellite-based approaches have been developed to measure changes in glacier mass
(Taylor et al., 2021; Bamber and Rivera, 2007; Berthier et al., 2023). These approaches include digital elevation model
(DEM) differencing from stereo-imagery and synthetic aperture radar interferometry, repeat radar and laser altimetry, and
space gravimetry. These approaches have recently been extensively applied to maritime glaciers within the southeastern
Tibet, enabling the acquisition of glacier mass balance data at various temporal and spatial scales (Wang et al., 2021; Yi et



al., 2020; Zhou et al., 2018; Wang et al., 2018). The conventional method of using altimetry satellites to ascertain the elevation of glacier surfaces for the purpose of calculating glacier mass change typically utilizes a single glacier boundary. This approach, however, fails to consider the contribution of dynamic glacier area to glacier mass change.

The objectives of this study are: (1) to develop an automated approach for extracting glacier boundaries using multi-source
data on the GEE platform; (2) to generate annual glacier boundary for southeastern Tibet; and (3) to analyse glacier changes and glacier mass balance considering dynamic variations in glacier area.

## 2 Study area and data

### 2.1 Study area

Southeast Tibet lies in the southeastern part of the Tibetan Plateau, with latitudes ranging from 27° to 32°N and longitudes
from 91° to 99°E, as shown in Figure 1 (Li Xingdong, 2022; Zhao et al., 2022), The region is at the junction of the Tibetan Plateau and the Yunnan-Tibet Plateau, including Tibet's Nyingchi, Sichuan's Garzê, Yunnan's Nujiang, Diqing, and Tibet's Chamdo Prefectures. The total area of the region is about 152,000 square kilometers.

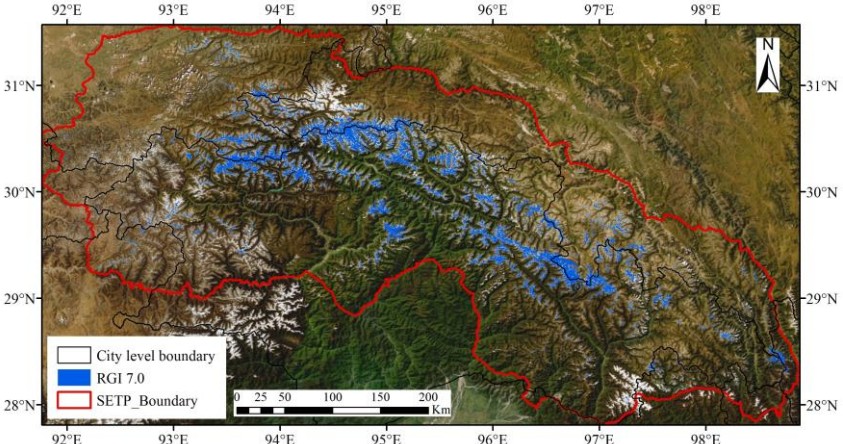

**Figure 1 Study area, the image presents remote sensing data, including the study area, city boundaries, and the RGI 7.0 glacier**
**inventor (Consortium, 2023)**

Southeast Tibet is located in a plateau area with complex topography and overlapping mountains. Mountain peaks generally exceed 5,000 meters, with rivers deeply cutting through gorges, resulting in significant elevation variations.

Southeast Tibet is an important glacier distribution area in China, with abundant glacier resources. According to the second glacier inventory of China, the study area has a glacier area of approximately 10,000 square kilometers, comprising 6,712
glaciers, with a total ice volume of 923.84 km³. Glacier area classification statistics show that in the southeastern Tibet, 4,935 glaciers have an area of less than 1 km², accounting for 73.5% of the total number of glaciers; 1,623 glaciers have an area between 1 and 10 km², accounting for 24.2%, and 154 glaciers have an area greater than 10 km², making up only 2.3%





of the total number of glaciers. Most of the glaciers in southeastern Tibet are located between high mountain gorges and peaks, with elevations generally ranging from 4,000 to 6,000 meters. Southeastern Tibet is also a significant marine-type

glacier accumulation zone, characterized by high accumulation and high ablation rates, making it particularly sensitive to climate change. Due to recent warming and reduced monsoon precipitation, glaciers in this region have accelerated their retreat since the 1990s, with a negative mass balance (Yang et al., 2010a; Sakai and Fujita, 2017).

## 2.2 Data

### 2.2.1 Landsat

This study utilizes the Landsat-5, Landsat-7 (pre-2003), and Landsat-8 Surface Reflectance Tier 1 datasets, which are available in an analysis-ready format on GEE. These datasets provide reflectance data processed through calibration and atmospheric correction, enhancing the representation of surface features. They include visible (VIS; 400–700 nm), near-infrared (NIR; 700–900 nm), and shortwave infrared (SWIR; 1400–2400 nm) bands, all with a spatial resolution of 30 m. Despite the 16-day revisit period of Landsat, image quality may be affected by cloud cover, seasonal snow, and sensor

anomalies. Therefore, careful image selection is required. The primary method involves using all valid (Cloud score less than 60) images from June to November within the study period on the GEE for statistical analysis and generating annual composite images. This approach mitigates the effects of cloud cover and seasonal snow on the results.

### 2.2.2 Sentinel-2

The Sentinel-2 mission consists of two polar-orbiting satellites in a single orbit: Sentinel-2A, launched in June 2015, and

Sentinel-2B, which became operational in July 2017. The mission offers a wide coverage and a short revisit period, with data collected every 2–5 days. Both satellites are equipped with advanced multispectral sensors that provides 13 spectral bands, ranging from visible light and near-infrared to shortwave infrared, with varying spatial resolutions. This study uses data from the Sentinel-2 Surface Reflectance Tier 1 dataset, available in an analysis-ready format on GEE.

### 2.3.3 Sentinel-1

The Sentinel-1 satellite is a synthetic aperture radar (SAR) mission launched by the European Space Agency (ESA). This study utilizes the COPERNICUS/S1_GRD dataset, accessed through the GEE platform. The dataset offers high temporal resolution, with observation intervals of six days. The VV band, representing vertical-vertical polarization, offers high resolution and multi-temporal observations. The data remains reliable during precipitation and cloud cover, which can otherwise degrade data quality. In this study, the VV band data is used as supplementary data for training the classification

model, thereby fully leveraging its resilience to cloud cover and other environmental obstructions.



### 2.2.4 NASADEM

The National Aeronautics and Space Administration (NASA) released the National Aeronautics and Space Administration Digital Elevation Database (NASADEM), a high-resolution digital elevation model (DEM) created by integrating satellite, aerial, and ground measurement data (Jpl, 2020). NASADEM has a spatial resolution of 30 meters, global coverage, and elevation accuracy within 1 meter. NASADEM integrates data from NASA's ICE-Sat and ASTER satellites, SRTM, and NGA. Advanced processing and fusion algorithms are used to integrate and refine these datasets. The data undergoes several processes to generate a high-quality DEM. These include global seamless stitching, removal of height influences from trees, buildings, and other objects, and correction of elevation offsets caused by water reflections. In this study, the NASADEM dataset is used to calculate elevation, slope, aspect, and shadow of mountain within the study area.

### 2.2.5 RGI 7.0

The Randolph Glacier Inventory version 7.0 (RGI 7.0) is a globally consistent dataset of glacier outlines (≥ 0.01 km²), excluding the Greenland and Antarctic ice sheets. Developed under the GLIMS initiative, RGI 7.0 provides standardized glacier geometries, enabling large-scale analyses of glacier distribution, dynamics, and climate sensitivity. For this study, the glacier inventory of the study area was derived from GGI18, which was developed under the Glacier Area Mapping for Discharge from the Asian Mountains (GAMDAM) project as an updated and refined version of the earlier CGI15 inventory (Nuimura et al., 2015; Sakai, 2019).

## 3 Methods

The random forest algorithm was used to automatically identify glaciers by integrating spectral, topographic, texture, and radar data. The workflow, outlined in Figure 2, includes data preprocessing, feature extraction, feature analysis, random forest classification, and result merging. For the years 2000, 2005, 2010, and 2015, the methodology was slightly adjusted due to the absence of Sentinel data. In these cases, the model was constructed solely using Landsat and NASADEM data. Additionally, the input data were not restricted to the target year but also incorporated images from the preceding and following years to enhance robustness. Notably, we constrain the results using the RGI 7.0 bounds.



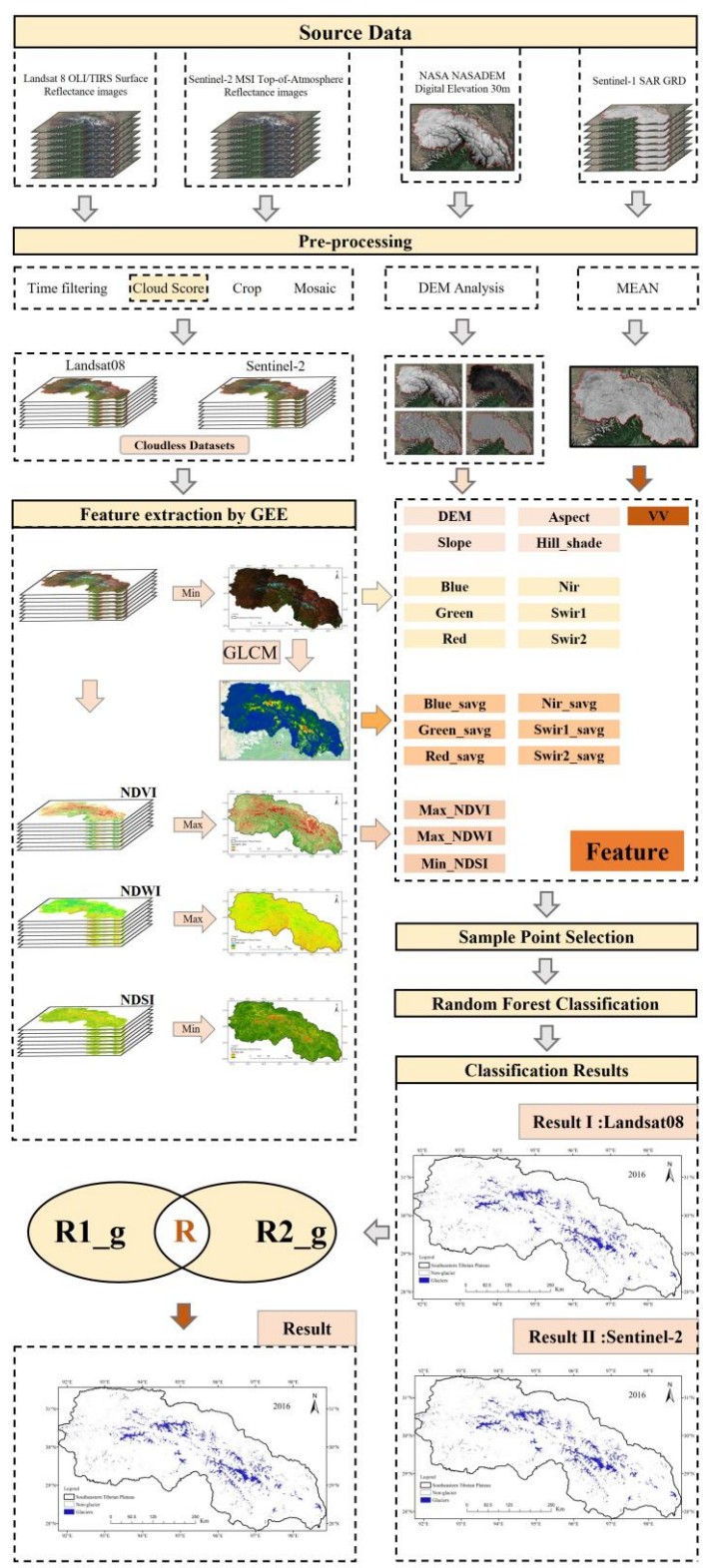



**Figure 2 The workflow chart, which begins with the integration of Landsat 8 and Sentinel-2 data (Sentinel series data not introduced until after 2016), combined with NASADEM and Sentinel-1 data, following a series of data pre-processing steps. Twenty features—including spectral, texture, index, radar, topographic attributes, and sketching samples—are extracted to construct a random forest classifier. This results in the generation of two groups of glacier extraction results at the decision level. The final results are achieved through the amalgamation of these results.**

## 3.1 Data Preprocessing

This study used all available images from June to November to generate composite images with minimal cloud and snow cover. This approach minimizes the influence of environmental factors on glacier extraction, ensuring more accurate and reliable glacier extraction. The preprocessing phase consists of three main stages: dataset selection, cloud filtering, and image synthesis. Dataset selection ensures only the most relevant summer images are included, cloud filtering removes cloud-affected regions, and image synthesis combines cloud-free images into a final composite.

### 3.1.1 Dataset Selection

The distinctive geographical position and climatic attributes of southeastern Tibet give rise to a considerable prevalence of cloud cover during the summer. After the glacier ablation season, the glacier and surrounding areas are least affected by snow. This study used all available imagery from Landsat-5/7/8, Sentinel-1, and Sentinel-2 data collected in the study area between June and November.

### 3.1.2 Cloud Filtering

The cloud removal algorithm (Gorelick et al., 2017) uses a scoring system to quantify cloud content in each pixel. This system integrates three key variables: brightness, temperature, and the Normalized Difference Snow Index (NDSI). This method prevents erroneous removal of clean ice or snow and optimizes cloud removal performance in high-altitude, cold regions (Xie et al., 2022).

The cloud score for each pixel is calculated using the cloud removal algorithm, ranging from 0 (no clouds) to 100 (dense clouds). The GEE cloud scoring algorithm computes several indicators, including brightness in the blue band, presence in both visible and infrared bands, low temperatures, and distinction from snow using the NDSI. Eq. (1) is employed to rescale the cloud indicators, normalizing the image bands (blue, red, near-infrared, temperature, and NDSI) with an upper threshold (Threshold1) and a lower threshold (Threshold0). The upper and lower thresholds are provided by GEE, as illustrated in Eq. (2). The minimum rescaled value is employed as the cloud score for each pixel, as illustrated in Eq. (3).

$$band_{re} = rescale(band, [threshold0, threshold1]) = \frac{band - threshold0}{threshold1 - threshold0} \qquad (1)$$

$$\begin{cases} blue_{re} = rescale(\rho_{Blue}[0.1, 0.3]) \\ visible_{re} = rescale(\rho_{Red} + \rho_{Green} + \rho_{Blue}[0.2, 0.8]) \\ infared_{re} = rescale(\rho_{Nir} + \rho_{Swir1} + \rho_{Swir2}[0.3, 0.8]) \\ temp_{re} = rescale(\rho_{Temp}[300, 290]) \\ NDSI_{re} = rescale(NDSI[0.8, 0.6]) \end{cases} \qquad (2)$$



$$cloudscore = minimum(blue_{re}, visible_{re}, infared_{re}, temp_{re}, NDSI_{re}) * 100 \quad\quad (3)$$

The cloud score for each pixel is calculated, and based on experiments, a threshold of 60 is set. Pixels with score above 60 are excluded, removing cloud cover while retaining the maximum usable data.

### 3.1.3 Image Synthesis

Landsat 5/7/8 and Sentinel-2 images were processed and corrected on GEE. After applying cloud filtering, a cloud-free pixel set is generated for each pixel location. Data cubes are then constructed by integrating optical and radar images.

## 3.2 Feature Extraction

Only spectral reflectance is insufficient for glacier identification. supplementary features are essential to improve classification precision. This study extracts spectral features, terrain, texture, and radar interferometric features to develop a

Random Forest classifier. The combination of these features improves the classifier's ability to distinguish glaciers from other landscape.

### 3.2.1 Optical Features

(1) Reflectance Features

Cloud-filtered, cloud-free pixel datasets from Landsat and Sentinel-2 are used to generate minimum value composites,

producing two annual cloud-free composite image sets, as illustrated in Figure 5 (a)-(f). These composites represent the minimum reflectance values for each pixel throughout the year, reducing the effects of cloud cover and atmospheric disturbances. Six spectral reflectance features, including all visible light bands, near-infrared (NIR), and shortwave infrared (SWIR), are selected for analysis. These spectral features are crucial for distinguishing different surface types, with glaciers serving as a prime example, displaying distinct reflectance characteristics compared to the surrounding terrain. To ensure

spatial resolution consistency, all datasets are resampled to 30 meters resolution.

(2) Spectral Index Features

To enhance the accurate classification of surface types, this study selects commonly used spectral indices, including the Normalized Difference Vegetation Index (NDVI), Normalized Difference Water Index (NDWI), and Normalized Difference Snow Index (NDSI). The definitions and calculation formulas for these indices are detailed in the following section.

**Normalized Difference Vegetation Index (NDVI)**

The Normalized Difference Vegetation Index (NDVI) is commonly used for evaluating vegetation coverage on Earth's surface (Rouse Jr et al., 1974). NDVI is calculated using the following formula:

$$NDVI = \frac{\rho_{NIR} - \rho_{Red}}{\rho_{NIR} + \rho_{Red}} \quad\quad (4)$$

In this formula, $\rho_{NIR}$ represents the reflectance value of the near-infrared band, while $\rho_{Red}$ denotes the reflectance value of the red band. The NDVI ranges from -1 to 1, with higher values indicating more abundant vegetation coverage. In this study,



the cloud-free pixel sets, obtained after cloud filtering, serve as the data source for calculating NDVI. NDVI is calculated on
a pixel-by-pixel basis, producing a cloud-free NDVI image set for each year from 2000, 2005, 2010, 2015, and 2016 to 2022.

**Normalized Difference Water Index (NDWI)**

The Normalized Difference Water Index (NDWI) is a common index for evaluating the distribution and extent of surface
water bodies (Mcfeeters, 1996). NDWI is calculated using the following formula:

$$NDWI = \frac{\rho_{Green} - \rho_{NIR}}{\rho_{Green} + \rho_{NIR}} \tag{5}$$

In this formula, $\rho_{NIR}$ represents the reflectance value of the near-*band, and $\rho_{Green}$ represents the reflectance value of the
green band. The NDWI value ranges from -1 to 1. A value close to -1 indicates the presence of pure water bodies, while a
value near 0 suggests a moderate distribution of surface water. A value close to 1 indicates minimal or no surface water. In
this study, NDWI is calculated on a pixel-by-pixel basis, producing a cloud-free NDWI image set for each year from 2000,
2005, 2010, 2015, and 2016 to 2022.

**Normalized Difference Snow Index (NDSI)**

The Normalized Difference Snow Index (NDSI) is used for detecting snow cover on Earth's surface (Hall et al., 1995), using
the following formula:

$$NDSI = \frac{\rho_{Green} - \rho_{SWIR1}}{\rho_{Green} + \rho_{SWIR1}} \tag{6}$$

In this formula, $\rho_{Green}$ represents the reflectance value of the green band, and $\rho_{SWIR1}$ represents the reflectance value of the
short-wave infrared band. The NDSI value ranges from -1 to 1, with higher values indicating more extensive snow cover. A
value is close to 1 indicates pure snow cover, while a value near 0 suggests minimal or no snow cover; A value close to -1
indicates non-snow materials. NDSI is calculated on a pixel-by-pixel basis, producing a cloud-free NDWI image set for each
year from 2000, 2005, 2010, 2015, and 2016 to 2022.

(3) Spectral Index Composition

Constructing spectral index features is shown in Figure 3. This involves processing cloud-free image datasets for the
Normalized Difference Vegetation Index (NDVI), Normalized Difference Water Index (NDWI), and Normalized Difference
Snow Index (NDSI), which were previously obtained.



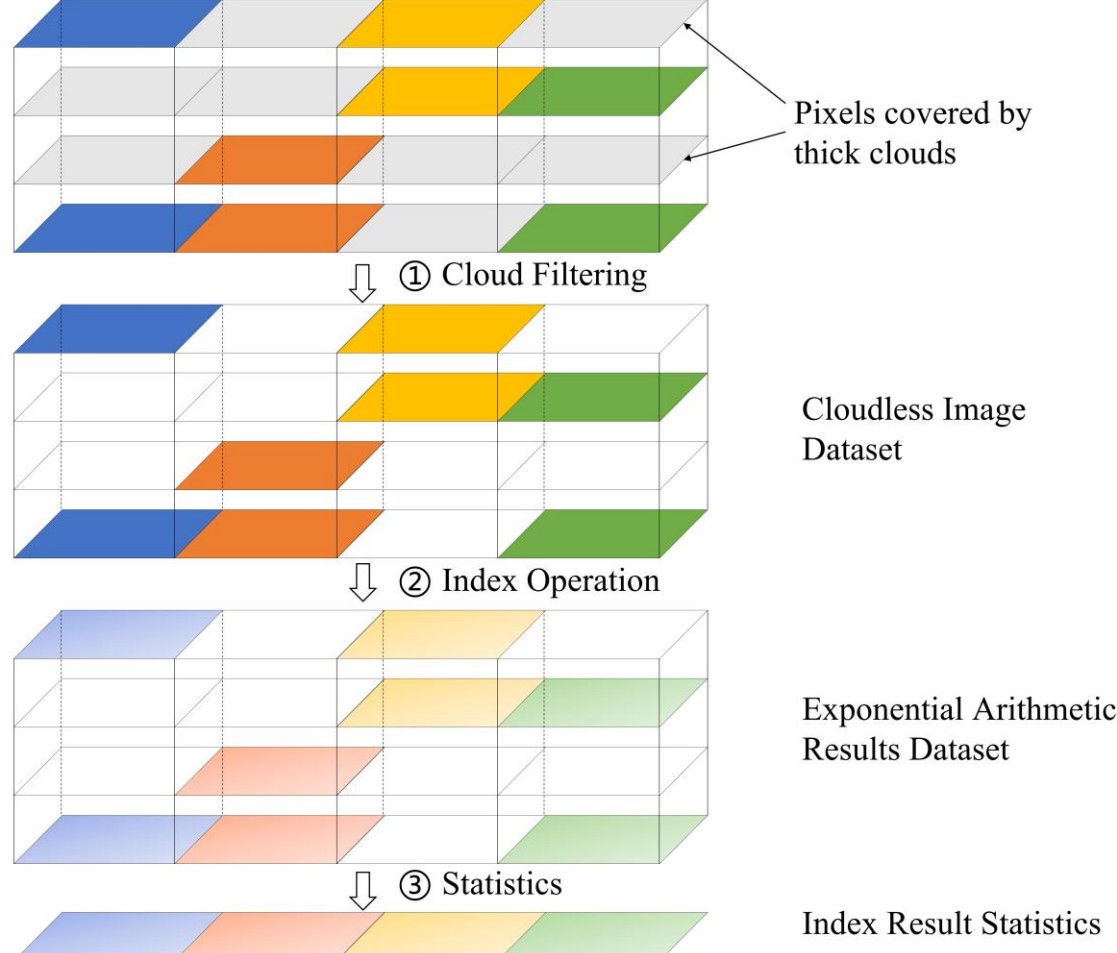

**Figure 3 Annual index synthesis methodology. Cloud removal, index calculation, and annual index synthesis on a pixel-by-pixel.**

The methodology for processing each pixel is consistent across all years. Firstly, a maximum value composition is applied to
the NDVI image set to derive the annual maximum NDVI (MaxNDVI) image. The image represents the peak vegetation
cover for each year. Similarly, a maximum value composition is applied to the NDWI image set to obtain the annual
maximum NDWI (MaxNDWI) image, reflecting the largest extent of water bodies each year. This step ensures accurate
documentation of water coverage variations. To generate the annual minimum NDSI (MinNDSI) image, a minimum value
composition is applied to the NDSI image set. This generates a representation of the lowest snow cover for the year,
minimizing the impact of seasonal snow variations on subsequent analysis. This systematic approach guarantees accurate
representation of key features of vegetation, water, and snow cover in the annual composite images. By focusing on
maximum and minimum values, we highlight the key characteristics of each land cover type throughout the year.



As illustrated in Figure 4, selected areas display the results of spectral index composition. The index composition could clearly distinguish between different land cover types so that provide a more accurate representation of glacier distribution.

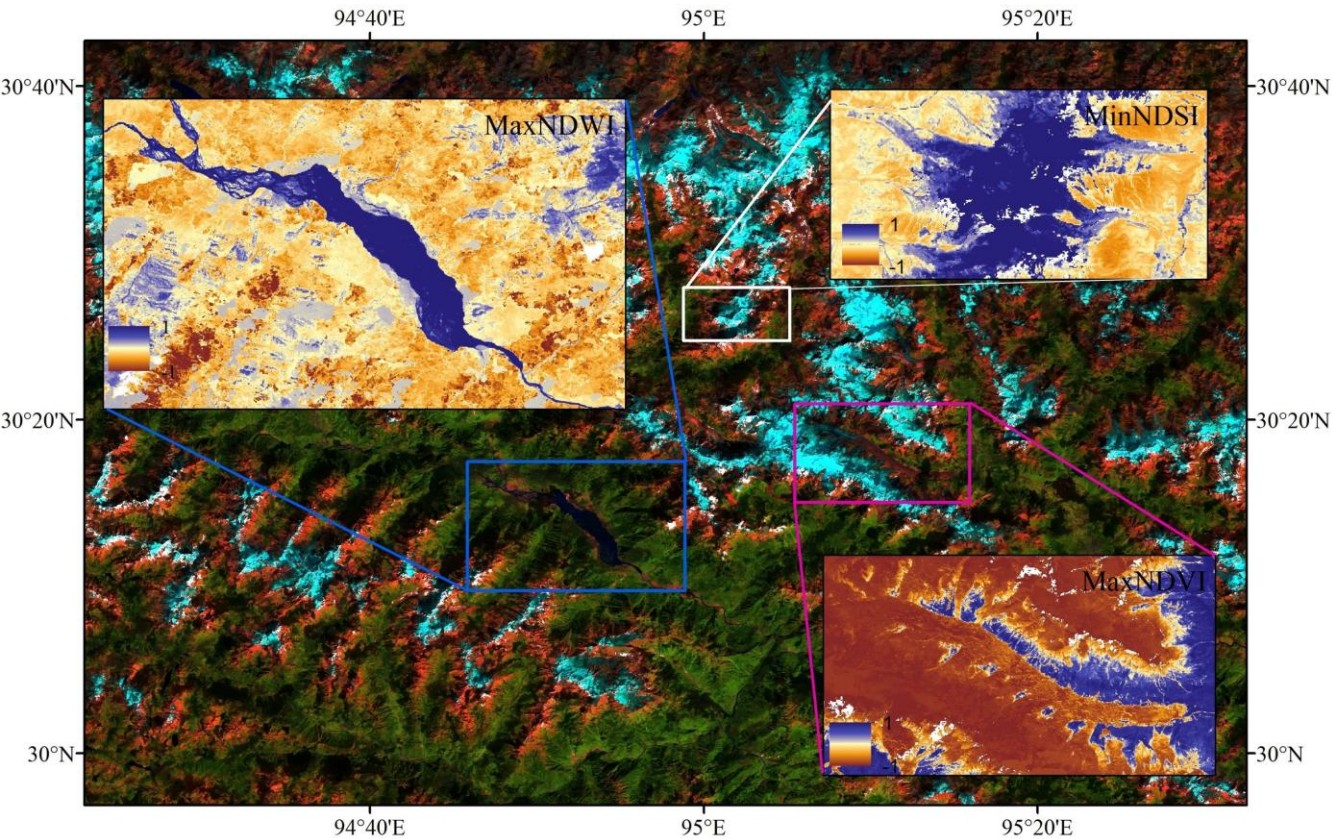

**Figure 4 Comparison of annual index results, MaxNDVI and MaxNDWI were obtained from the annual maximum normalized vegetation index and the annual maximum normalized water index, respectively.**

### 3.2.2 Texture Features

Only using spectral features for glacier extraction may not be sufficient. Texture features are characteristics of an image that are independent of color or brightness and capture the visual patterns of uniformity. These features provide essential information about the structure, configuration, and interactions of objects with their environment. The gray-level co-occurrence matrix (GLCM), proposed by Haralick (Haralick, 1979), is a widely used method for texture description, examining the spatial correlation of gray levels.

In this study, the GEE provides the GLCM-Texture function, enabling efficient computation of texture features based on the GLCM. The texture features, including mean, angular second moment, contrast, correlation, variance, dissimilarity, homogeneity, and entropy (Khan et al., 2020), were calculated pixel-by-pixel using the annual cloud-free minimum value composite images.



Texture features were extracted from six bands—visible light, near-infrared, and short-wave infrared—of the annual minimum value composite images for each year from 2000, 2005, 2010, 2015, and 2016 to 2022, resulting in 48 images. In a

recent study, Y. Lu et al.(Lu et al., 2020) conducted a correlation analysis of eight texture indices and found that the mean value is consistent with other texture features, playing a key role in glacier mapping. To enhance the accuracy of glacier classification, this study selected the mean texture feature derived from the six bands as supplementary data to support the automated delineation of glacier boundaries.

### 3.2.3 Topographic Features

Topographic features are described using parameters, such as elevation, slope, aspect, terrain curvature, and roughness. This study selected four topographic indicators—elevation, slope, aspect, and terrain shadow—for classification analysis based on NASADEM. These indicators were chosen for their relevance to the classification task and their ability to represent key landscape features.

### 3.2.4 Radar Image Features

The VV band of Sentinel-1 imagery was also used for training the random forest classifier. The VV band represents vertical-vertical polarization, where the Synthetic Aperture Radar (SAR) transmits and receives vertically polarized microwave signals. A mean synthesis was applied to the annual Sentinel-1 images pixel-by-pixel to reduce salt-and-pepper noise.

### 3.3 Feature Analysis

A total of 20 features were used as input for the random forest classifier. including optical features (blue, green, red, near-

infrared, shortwave infrared 1 and 2, MaxNDVI, MaxNDWI, MinNDSI), texture features (mean texture of spectral reflectance for the six bands), terrain features (elevation, slope, aspect, and terrain shadow), and radar features (VV polarization band). As shown in Figure 5, the surface reflectance characteristics in panels (A) through (F) reveal substantial overlap in reflectance information for pristine glaciers, debris-covered glaciers, and other land cover types. The mean texture feature images in panels(G)-(L) outline the general contours of the glaciers. The MaxNDVI index in panel (M) effectively

distinguishes high NDVI values indicative of vegetation cover from low or negative values of other land cover types. Similarly, the MaxNDWI feature in panel (N) effectively differentiates glaciers from water bodies. The NDSI index has long been a key tool for glacier extraction. The MinNDSI in panel (O) provides a more accurate estimate of the minimum glacier area for each year, reducing the impact of seasonal snow on the extraction process.

As shown in Figure 5 panel (P) - (S), the terrain features indicate that bare glaciers and debris-covered glaciers have higher

elevations than other land covers. Water bodies and debris-covered glaciers have relatively gentle slopes, while mountain and glacier shadows share consistent aspect characteristics, with high terrain shadow values for both. The radar VV polarization band image in panel (T) distinguishes bare soil from debris-covered glaciers, as the thin debris layer allows the



VV band penetration, detecting reflections from the underlying ice surface and revealing the glacier reflectance characteristics.





**Figure 5 A total of 20 features were used as input. (A) - (F) represent spectral features (corresponding to red, green, blue, near-infrared, shortwave infrared band 1, and shortwave infrared band 2, respectively); (G) - (L) correspond to texture features (mean texture features for red, green, blue, near-infrared, shortwave infrared band 1, and shortwave infrared band 2, respectively); (M) denotes the annual maximum Normalized Difference Vegetation Index (MaxNDVI); (N) represents the annual maximum Normalized Difference Water Index (MaxNDWI); (O) refers to the annual minimum Normalized Difference Snow Index (MinNDSI); (P) - (S) represent topographic features (elevation, slope, aspect, and mountain shadow, respectively); (T) corresponds to the Sentinel-1 radar VV polarization band; (U) shows the classification results from the random forest classifier.**

## 3.4 Random Forest Algorithm

Random forest is an ensemble method based on decision trees, where multiple trees are combined, and the final classification or prediction is determined by majority voting (Breiman, 2001). The random forest method has demonstrated strong performance in various practical applications, including classification, regression, and feature selection. This is attributed to its ability to handle high-dimensional data and to identify complex patterns. Extensive research has demonstrated the superiority of the random forest algorithm in classification, especially when using multi-source data and analyzing it from diverse perspectives (Alifu et al., 2020; Olthof and Rainville, 2022; Kulinan et al., 2024).

### 3.4.1 Classification System Determination

The region is classified into six land cover types: clean glacier, debris-covered glacier, bare land, water bodies, vegetation, and mountain shadows. The classification labels, manually interpreted, are shown in Figure 6.



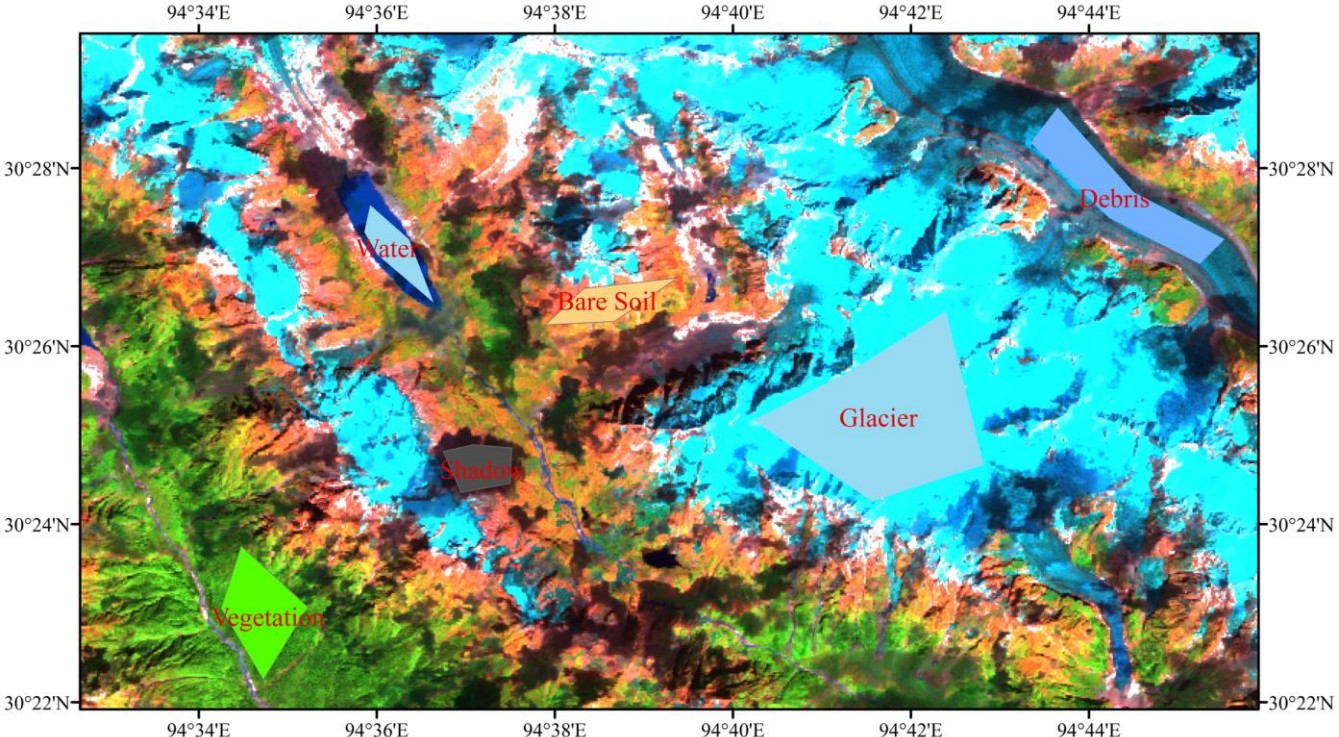

**Figure 6 Manual interpretation of labels, Six types of features (clean glaciet, debris-covered glacier, bare land, water bodies, vegetation, and mountain shadows) and their artefactual interpretation markers.**

### 3.4.2 Selection of Classification Samples

We manually interpret and select 200 sample points for each land cover types: clean glacier, debris-covered glacier, bare land, water bodies, vegetation, and mountain shadows on GEE. The sample points were selected to ensure a representative distribution of land cover types across the study area. In total, 21600 sample points were selected, covering an area of 19.44 km². These points represent a diverse range of land cover types, providing a comprehensive dataset for analyzing glacier and land cover dynamics in southeastern Tibet.

The calculated Jeffries-Matusita (J-M) distances showed that all selected training samples had a J-M distance greater than 1.8, indicating high separability between the land cover types in the dataset. For each year's samples, 70% were used for training the random forest classifier, with the remaining 30% were reserved for accuracy validation. This split ensures a robust evaluation of model performance. The training and testing datasets are randomly split according to a predefined ratio in each iteration to ensure the model's robustness and generalizability.

### 3.4.3 Classification Accuracy Evaluation

In this study, the confusion matrix method was used to assess the accuracy of the classification model(Townsend, 1971). A confusion matrix was constructed to calculate several accuracy metrics, including overall classification accuracy, the Kappa



coefficient for overall performance, and producer's and user's accuracy for evaluating the performance of individual

categories. The metrics provided comprehensive insights into both the overall classification results and the specific

categories performance.

### 3.5 RGI 7.0 Boundary Constraints

We utilized the RGI 7.0 glacier boundary dataset to constrain our automated glacier mapping workflow to manually validate

boundaries around the year 2000, aiming to reduce the misclassification of seasonal snow cover in later periods. This

constraint introduces an inherent assumption that all glaciers in the study area have retreated from their 2000 extents.

Additionally, it inherits the uncertainties and omissions associated with the RGI 7.0 dataset. No existing studies have

reported any glacier advances in the southeastern Tibetan Plateau between 2000 and 2022.

### 3.6 Combined results of Landsat-8 and Sentinel-2 images and pixel complementation

Because of the introduction of the Sentinel data, we realized year-by-year glacier extraction in 2016-2022 using Landsat and

Sentinel data sets to construct models and classify them separately. This study produced two sets of classification results,

using Landsat-8 and Sentinel-2 imagery along with DEM and Sentinel-1 data. Due to missing values in the images after

cloud filtering, the two sets of results were merged at the pixel level using the following rule: a pixel is classified as glacier if

both sets identify it as such, or if one set identifies it as glacier while the other is missing; all other cases are classified as

non-glacier. This approach was chosen to minimize the impact of missing data and improve the accuracy of the final

classification results.

Despite using multiple techniques to avoid misclassifying snow as glaciers, the impact of seasonal snow still led to some

misclassifications. To address this issue, a decision-level fusion strategy was applied to reduces the impact of seasonal snow

on the final glacier extraction results.

There are still less than 1% missing pixels of the study area in final annual glacier boundary. These missing pixels are often

concentrated near glaciers. This concentration may disproportionately affect the analysis of glacier dynamics and area

changes. This phenomenon may impact the interannual variation in glacier area, especially near glacier termini. To address

this issue and enhance the spatial completeness of the dataset, this study incorporated the RGI7.0 glacier inventory to

directly fill in the missing pixels.

### 3.7 Post-Classification Processing

Pixel-based classification often results in isolated pixels and noise. To mitigate small-area noise and enhance spatial

consistency, this study employs ArcGIS Focal Statistics for post-processing. Specifically, a 5×5 majority filter is applied,

where each pixel is reassigned the most frequently occurring class within its 5×5 neighborhood. This approach effectively

removes isolated pixels, improves the spatial continuity of homogeneous land cover types, and preserves the integrity of

major land cover boundaries.



# 4 Results and analysis

## 4.1 Glacier extraction results and error analysis

The annual classification results were evaluated using the confusion matrix method, showing high accuracy with Kappa coefficients above 93% and overall accuracy exceeding 94%. These results highlight the robustness and reliability of the
classification approach for glacier extraction.

We provide a clear visual representation of specific glacier for detailed comparison in the Figure 7, which shows the 2000 annual glacier boundary and RGI 7.0 statistical result. Both clean glaciers and surface moraine-type glaciers are well identified, but for some missed glacier areas, this is mainly due to shadows obscuring the glacier. Extracts for all years are shown in the supplementary document.

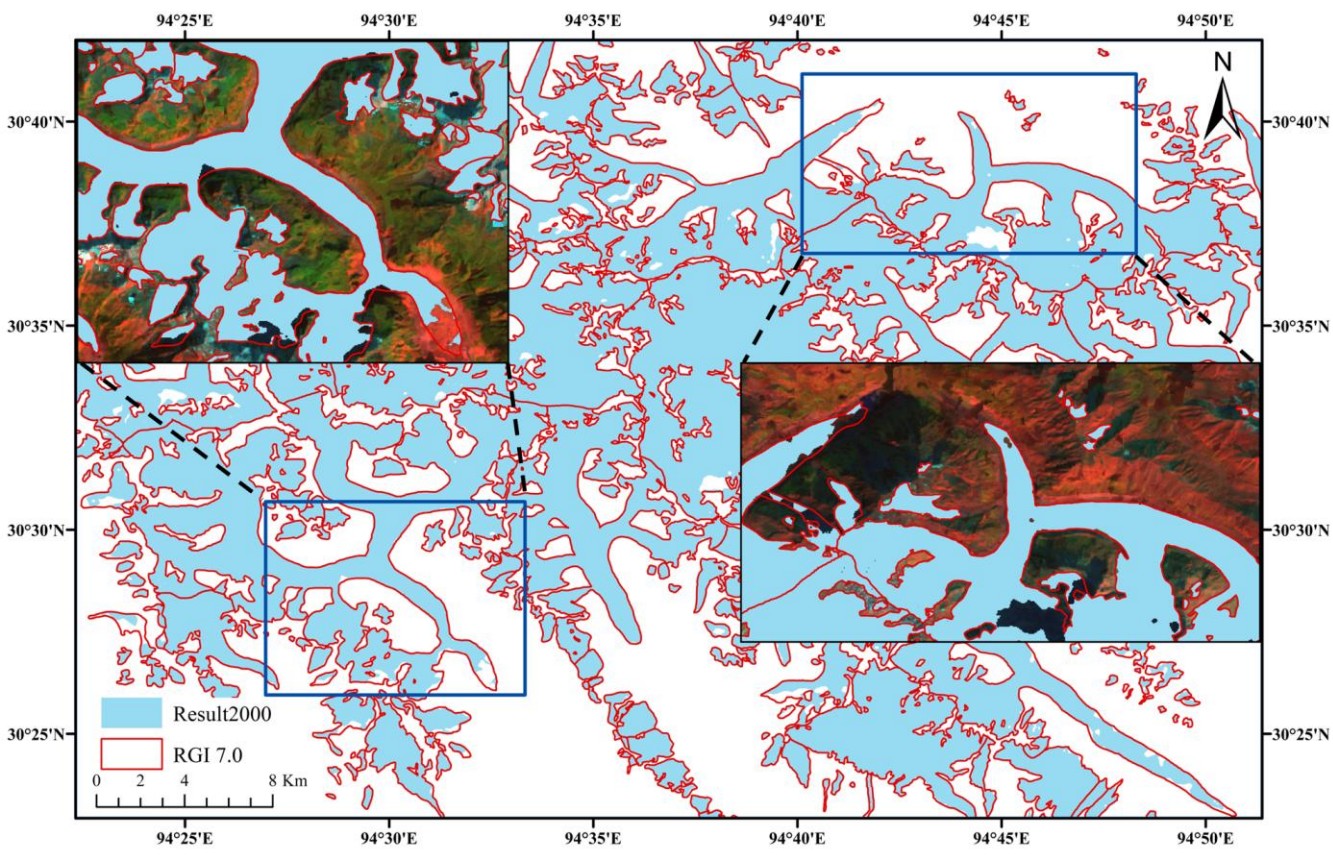


**Figure 7 Glacier boundary at 2000 and RGI7.0 glacier boundary.**

## 4.2 Glacier area change in Southeast Tibet

Our glacier area included both clean and debris-covered glaciers. A clear pattern of glacial retreat is evident at the terminus, indicating a continuous reduction in ice mass in the region. Changes in glacier extent from 2000, 2005, 2010, 2015, 2016 to





2022 were calculated based on these estimated areas and are shown in Figure 8. The calculation of the mapping error was based on half an image element (15 m) (Paul et al., 2013). Overall, the glaciers in the southeastern Tibet region experienced a consistent decline from $7898.61 \pm 652.15$ km$^2$ in 2000 to $6317.13 \pm 592.57$ km$^2$ in 2022. Excluding mapping errors, the glaciers are retreating at a rate of $85.03 \pm 7.60$ km$^2$/y. Notably, the retreat rate accelerated after 2010, increasing from $57.72 \pm 16.81$ km²/y (2000–2010) to $97.72 \pm 17.67$ km²/y (2010–2022).

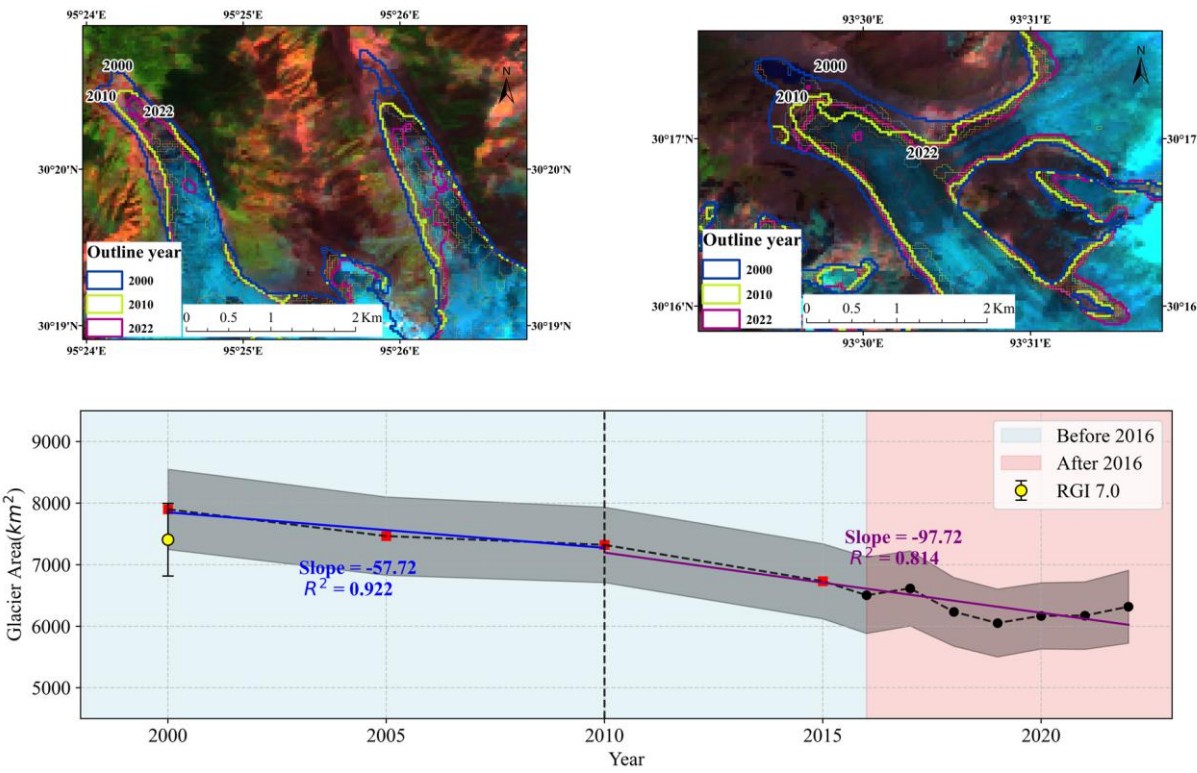


**Figure 8 The top picture shows the retreat of the glacier at the end of the tongue. the low show time series of total southeast Tibet glacier area changes over 2000-2022.**

## 5 Discussion

### 5.1 Compare with previous results

There are few annual observations of glacier area changes in southeastern Tibet. We compared with previous reports that cover our study regions and close time periods. The Chinese Second Glacier Inventory (V1.0), based on 2006-2011 data, estimated a glacier area of 10,288.47 km². in southeastern Tibet. it is note that some glacier areas in the First Glacier Inventory precede this range. Ye et al. (Qinghua, 2019, 2020)reported a glacier area of 9,870.47 km² (2013), with their 2017 estimate at 7,251.48 km²—both higher than the results of this study (6614.10 $\pm$ 609.43 km² in 2017). The main reason for

this is because of the difference in the timing of the data, they are using winter data, which can lead to snow being





misclassified as glaciers. Ke et al. (Ke et al., 2016), using data from 2011-2013, compiled the Southeastern Qinghai–Tibet Plateau Glacier Inventory (SEQTPGI), which estimated an area of 6,566 ± 197 km². However, differences in the study area boundaries contributed to the observed inconsistencies.

The glacier area in southeastern Tibet has been debated. Discrepancies in reported results can be attributed to several factors
in extraction methods, datasets, study periods, and definitions of glaciers.

## 5.2 Correlation of information on glacier elevation changes with information on area changes

Glacier area fluctuates with changes in glacial thickness. This study uses the result glacier thickness changes result from the ICESat, Cryosat-2 and ICESat-2 to examine the correlation between changes in glacier area and thickness. The ICESat, and Cryosat-2 results were obtained using the results of Jakob et al.(Jakob et al., 2021) and the ICESat-2 results were calculated
using the method of Wang et al.(Wang and Sun, 2022; Wang et al., 2017) As shown in Figure 9, the annual fluctuations in glacier thickness and area in southeastern Tibet from 2002 to 2022 exhibit a strong correlation (R² = 0.89), based on data points from 2005, 2010, 2015, and 2016–2022. This supports our understanding of glacial retreat characteristics and further validates the accuracy of the glacier extraction results.

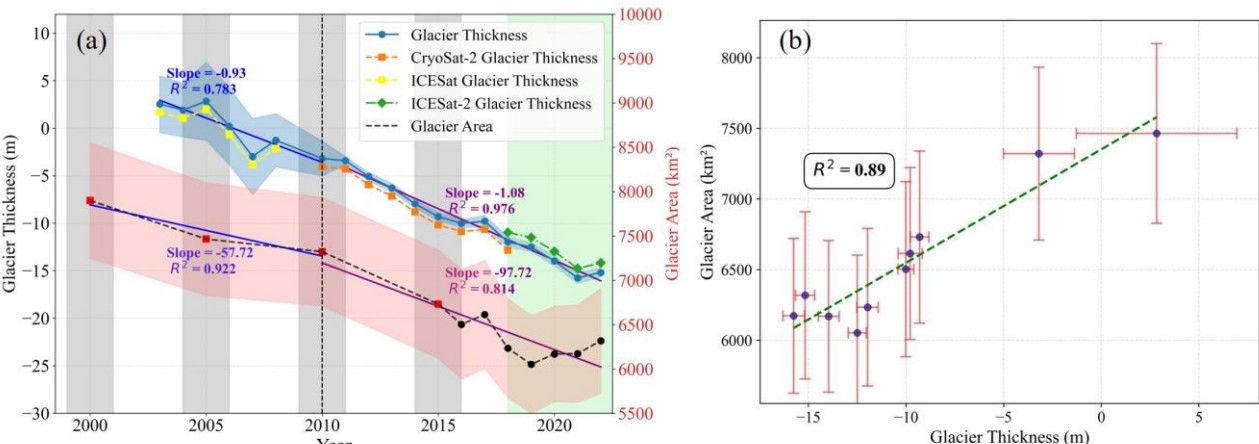

**Figure 9 Relationship between glacier area and thickness changes in Southeast Tibet, (a) Glacier thickness changes derived from three generations of altimetry satellites (ICESat, CryoSat-2, and ICESat-2). The data from different satellites were merged to produce a continuous thickness time series. For clarity, vertical offsets were applied during visualization. Light blue shading indicates the uncertainty in glacier thickness, and light red indicates the uncertainty in glacier area. (b) The correlation between changes in glacier surface elevation and its area.**

## 5.3 Glacier Mass Balance in Southeastern Tibet

This study combined year-to-year glacier area and glacier thickness change data to derive glacier mass change. Glacier mass change is annually by multiplying the average annual glacier area by the annual change in elevation, then by the ice density. The formula is:





$$\Delta M_i = S_i' \times \Delta H_i \times \rho_G \tag{7}$$

As the glacier area is variable, the average area for each year is calculated using the following formula(Zhang et al., 2013):

$$S_i' = \frac{1}{3}\left(S_{i-1} + S_i + \sqrt{S_{i-1}S_i}\right) \tag{8}$$

Using 2003 as the baseline year for calculations, let $S_{i-1}$ represent the initial glacier area (i.e., the area from the previous year), and $S_i$ represent the glacier area in the final state (i.e., the area for the current year). This allows us to derive the average glacier area $S_i'$ for the years 2004 to 2022. (Areas for 2003, 2004, 2006-2009, 2011-2014 we supplemented using linear interpolation) Substituting this value into the mass change formula yields the annual mass change of the glaciers in the southeastern Tibet relative to 2003. The glacier mass for 2003 is set to 0, producing the glacier change curve for 2003 to

2022, as shown in Figure 10. The results show that glaciers in southeastern Tibet are at a rate of 6.20 Gt/y. By incorporating dynamic area changes, this study provides high precision result of glacier mass balance.

To evaluate the impact of incorporating annual glacier inventories on mass change estimates, we also computed glacier mass loss using a fixed glacier area from 2003. The comparison shows that including annual inventories changes the estimated trend by over 10%, underscoring the importance of inventory updates for accurately capturing glacier mass changes.

We assessed uncertainties in glacier mass loss trends by considering three independent error sources: surface elevation, glacier area, and ice density. Glacier area uncertainty was derived from our inventory; although individual mapping errors were below 10%, we conservatively assigned a 15% uncertainty to account for glacier retreat and flow. Ice density was assumed to be 900 kg/m³ with a 5% uncertainty. Elevation uncertainty, reflecting both fitting and measurement errors, was estimated at 10%. Combining these uncertainties, the overall error in glacier mass loss was 18.7%, yielding a mass loss rate

of $6.20 \pm 1.16$ Gt/y.





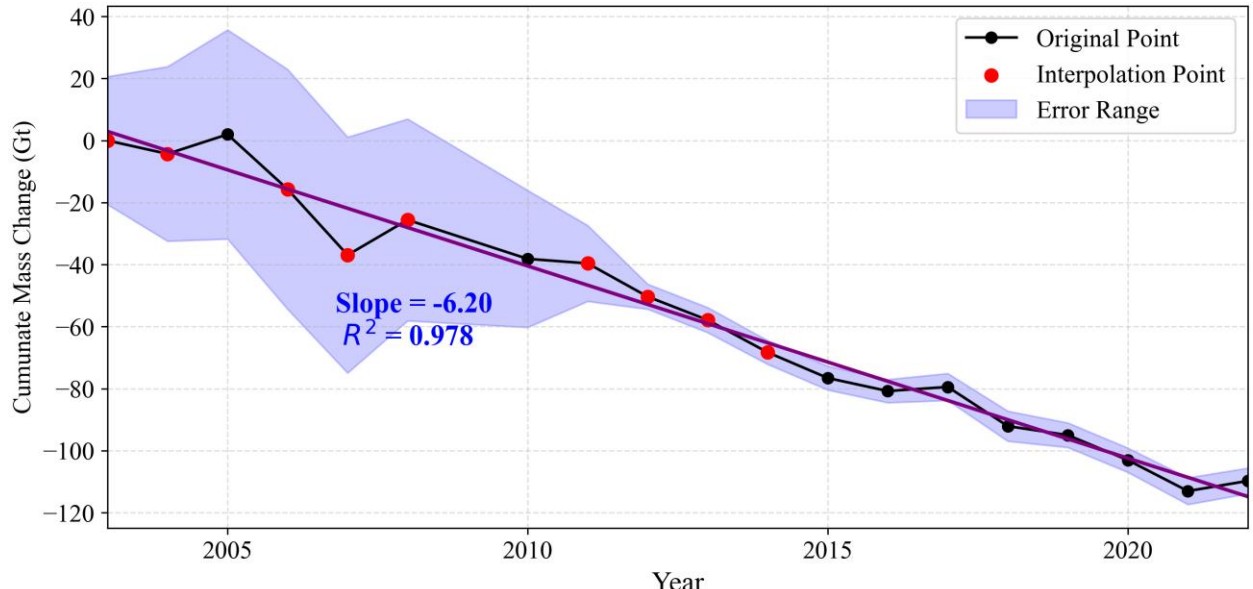

**Figure 10 Changes in glacier mass under year-to-year area changes**

As shown in Table 1, although the study periods and regions differ slightly, we compared our estimates with representative published results to better contextualize our findings. Due to differences in methodology, data sources, and study settings, some variation exists; however, the comparison shows that our estimates are consistent with and reasonable within the expected range.



**Table 1  Comparison of our results with previous studies.**

| Study | Data | Period | Regions similar to this study | Mass change rate (Gt/y) |
|---|---|---|---|---|
| (Brun et al., 2017) | ASTER stereo images | 2000-2016 | Nyainqentanglha | -4.0 ± 1.5 |
| (Shean et al., 2020) | WorldView/GeoEye DEMs, ASTER DEMs, and TanDEM-X Global DEM | 2000-2018 | Nyainqentanglha | -3.15± 0.93 |
| | | | Hengduanshan | -0.96 ± 0.23 |
| (Yi et al., 2020) | GRACE and ICESat | 2002-2017 | Nyainqentanglha | -6.5 ± 0.80 |
| (Wang et al., 2021) | ICESat-1,2 and GRACE/GRACE-FO | 2003-2019 | Nyainqentanglha | -6 ± 1.0 |
| (Jakob et al., 2021) | CryoSat-2 SARIn L1b | 2010-2019 | S AND E Tibet | -3.38 ± 1.21 |
| | | | Hengduanshan | -4.30 ± 0.98 |
| | | 2000-2019 | | -4.72 ±1.18 |
| | | 2003-2020 | | -4.86 |
| (Zhao et al., 2022) | ASTER stereo images, CryoSat-2 SARIn L2I, ICESat-1/2, and GRACE/GRACE-FO | | SETP | -5.80 ± 0.8(mean seasonal) |
| | | 2011-2020 | | -5.53 ± 0.2(mean annual) |
| This study | CryoSat-2, ICESat-1/2, Landsat-5/7/8, Sentinel-1/2 | 2003-2022 | * | 6.20 ± 1.16 |

## 5.4 Advantages and uncertainties

Glaciers are highly sensitive indicators of climate change, with changes in their area and mass widely used to assess climatic

responses at regional and global scales. However, repeated glacier inventories remain scarce, and no second comprehensive global inventory exists to date. The Randolph Glacier Inventory (RGI) currently available reflects glacier distributions circa 2000, limiting systematic analyses of glacier area changes over time. Most glacier mass change studies rely on a single static glacier boundary, focusing primarily on surface elevation changes while largely neglecting glacier area variations, which have not been quantitatively evaluated in detail. To fill this gap, we reconstructed glacier mass changes in southeastern Tibet

over the last two decades using a time series of annual glacier inventories. Our results reveal that glacier area changes contribute up to 10% of the total mass loss in this region, highlighting the potential bias introduced when ignoring area changes. This study is among the first to employ annually updated glacier inventories for mass change estimation, demonstrating their importance for improving accuracy and understanding glacier dynamics.

Nevertheless, this study has limitations, chiefly the following:

(1) Cloud cover:



A cloud score algorithm was applied during preprocessing to mask clouds, yet missing data remain. These gaps were filled using RGI 7.0. Missing values comprised less than 1% in single-date images and have minimal impact on results. However, cloud shadows are challenging to detect and may cause misclassification, especially over glaciers.

(2) Seasonal snow:

Despite efforts to minimize seasonal snow influence, some residual snow remains in composite images due to limited data availability. The slight area increase observed in 2020 is mainly attributed to seasonal snowpack. Incorporating additional data sources could mitigate this issue.

(3) Mountain shadows:

Accurately mapping glaciers in shaded alpine terrain remains difficult. While mountain shadows were classified separately
and terrain features were incorporated to reduce their impact, identifying glaciers in shadowed regions remains problematic. Observed gaps in ice tongues are primarily due to this limitation.

## 6 Conclusion

We use decades of features of satellite imagery, such as annual spectral indices, topographic features, texture, and radar band characteristics, to build a random forest classifier for glacier extraction. Using this method, we show for the first time the
year-to-year change of glacier area in southeastern Tibet during 2016-2022 while obtaining glacier inventories for 2000, 2005, 2010, and 2015. The total glacier area in the region decreased by $85.03 \pm 7.60$ km$^2$/y, and glacier retreat was found to have increased since 2010, increasing from $57.72 \pm 16.81$ km²/y (2000-2010) to $97.72 \pm 17.67$ km²/y (2010-2022). The correlation analysis between glacier thickness and area changes was also performed, showing a high correlation ($R^2 = 0.89$). Additionally, we optimized the glacier mass change estimates from altimetry data using dynamic annual area data, revealing
that glacier mass in southeastern Tibet is shrinking at a rate of $6.20 \pm 1.16$ Gt/y (2003-2022). This is the first use of dynamic annual glacier area changes to calculate glacier mass change, providing a new complement to altimetry-based glacier mass change studies. Practical applications show that our study could provide effective support for future glacier inventories and monitoring.

## Data Availability Statement

The data used in this study are open to the public. Landsat-8 image courtesy of the U.S. Geological Survey. Sentinel-1, Sentinel-2 data courtesy of ESA. NASADEM image courtesy of NASA JPL. These data were obtained and used online based on GEE (Google Earth Engine: https://earthengine.google.com). Our glacier extraction results (2000, 2005, 2010, 2015, 2016-2022) are available in the supplement of the paper.



## Acknowledgements

This study was financial supported by National Natural Science Foundation of China (42474007, 42104010, 42174097, 41974093, and 41774088), and the Fundamental Research Funds for the Central Universities.

## Declarations

Conflicts of interest: The authors declare no conflict of interest.

## Author contribution

Zihao Li and Qiuyu Wang designed the research. Zihao Li collected and processed the data. Zihao Li and Qiuyu Wang performed the analysis and interpreted the results. Wei Yang, Huan Xu and Wenke Sun assisted with the data interpretation and manuscript revision. Zihao Li wrote the initial draft. All authors contributed to manuscript editing and approved the final version.

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
