# Peer review of "Multi-Source Remote Sensing-Based Reconstruction of Glacier Mass Changes in Southeastern Tibet Since the 21st Century"

_EGUsphere, 2025_

## Referee Comment (RC2)

**Review of egusphere-2025-1772**

**General comments**

The study by Li et al. aims at presenting a method that allows the automated mapping of glacier extents in a challenging region using Google Earth Engine at high temporal resolution, up to annually from Sentinel-2. They used the results to consider area changes when calculating glacier mass balance. If the mapping had worked, this would have been an important study to improve related results also for other regions in the world. Unfortunately, the outcome of the mapping is not useful for any assessment. In this regard I want to acknowledge that the authors have provided the results of their mapping effort in the supplemental material. Without this, my evaluation would have been different as the paper is otherwise well written and the idea to just use all data available and combine it for the best possible result is fine. However, the largely arbitrary area changes from dataset to dataset (for individual glaciers partly larger than 50% from year to year) are obvious and glaciologically impossible. The authors mention that there are the usual problems with debris cover, clouds and shadow, but they have seemingly not recognized how large and arbitrary the variability is and that their method does not produce meaningful results.

Neither the unrealistic area increase by about 500 km$^2$ (estimated from Fig. 8, numbers for individual years are not provided) from 2019 to 2022 (naming it as a 'consistent decline' in L387), nor the sudden strong increase from 2016 to 2017 is discussed or considered as unrealistic. Instead, the authors correlate glacier elevation changes (wrongly labelled as 'Glacier Thickness' in Fig. 8) with glacier area changes as they assume there is a correlation (L407) and think that the correlation can be used as a validation (L413) of their (wrong) glacier areas. In fact, area changes are mostly driven by the ice thickness distribution along the glacier perimeter (thus depending on the shape of the glacier cross-profile) and are a longer-term response to changes in flow dynamics (glaciers have a response time). Hence, also the follow-on analysis is a bit strange. In this regard, it is also unclear to me why the authors rely on results from Cryosat and ICESat (with their diverse range of issues) for such small glaciers instead of the Hugonnet et al. (2021) dataset that is widely used? As this dataset is not even mentioned in the comparison Table 1, I wonder why. Is the dataset too bad in quality?

The very short results section (it has just 14 lines) mentioned the Kappa coefficient and overall accuracy along with three images showing outline overlays. I am aware that these statistical accuracy measures are frequently used in remote sensing studies to present the accuracy, but in my view they can be the result of anything and do not allow to obtain meaningful conclusions about the 'robustness and reliability of the classification approach'. At least for glaciers they do not work, as nicely confirmed here by the largely arbitrary results of the glacier mapping. The quality of mapped glacier extents can be shown by a) outline overlays and b) the sum of commission and omission errors (false positives and false negatives) divided by the common area. But as the former have been removed by the masking with RGI 7.0, I am unclear if the measures can be used here at all?

I see missing debris-covered glacier parts and that large regions in shadow are sometimes missing. Hence, intensive manual editing would be required before resulting outlines could be used for change assessment. The statement that glaciers 'are well identified' (L377) seems misleading in this regard. Although the year 2000 dataset is likely the most complete regarding shadow and debris mapping, showing a region where the method does not work and discussing it would have been more helpful. One can see the problems of the classification a little bit for the 2022 outlines of the right glacier in the upper left panel [please name them

properly a), b) and c)] of Fig. 8, but the image is very dark (what about some contrast stretching?), the lines are hard to see (also the red outlines on a reddish background in the insets of Fig. 7 are barely visible) and the wrong mapping results are not really discussed.

A final major point of concern is the general set-up of the study. First, the elevation change datasets are introduced in the Discussion Section 5.2 rather than in Sections 2.2 and 3. Their description is thus very short and the processing method unclear (e.g. how has the radar penetration into snow been corrected?). This can likely easily be adjusted. The motivation to determine annual area changes is mentioned, but not critically discussed. Even when the resulting glacier outlines would have been correct, a one-pixel (two pixels for debris-covered regions) uncertainty at 30 m resolution relates to a 30 and 60 m location uncertainty of the outline. With an assumed annual terminus retreat of 5 to 10 m / year (much less around the perimeter), one has to wait several years before new outlines make sense compared to uncertainties. But here the mapped termini could be wrong by several km, so change assessment is not an option. To make my major objection of the high variability in the mapping results clearer, I have added all glacier maps (setting the no data value to 0 before) and received a very colourful picture. On the last two pages of this review, I show a few examples for illustration. If the mapping had been correct, colours should only appear near the terminus and around the perimeter. As a note, this is just the result of a simple adding without a timeline, not revealing the partly strong year-to-year jumps in mapped glacier areas.

In conclusion, this brute-force mapping using sophisticated image processing without a sufficient understanding of the mapped subject (glaciers) and how it should change over time is not recommended. When being harsh, I would ask the authors to please first learn the basics about glaciers and how they work, then proceed with the user needs (are annual updates really required?) and then do the mapping. A bit less harsh I would ask the authors to first get the mapping right for one year before applying it to several years. When glacier area changes are mostly due to changes in the mapping results rather than real changes, there is no need to perform change assessment. In my view, it is possible to publish a study revealing that a method has not worked. However, in this case an honest discussion and illustration of the problems is required to be helpful for future studies. Concluding that this study provides 'effective support for future glacier inventories' (L482) is in my view highly misleading.

**Specific comments**
I do not comment here on all details of the study, but include some general remarks:
Providing area changes in $km^2$ and mass changes in Gt (sure to use 900 rather than 850 $kg/m^3$ in this region?) are not useful as they are incomparable across regions. In future studies, please give relative area changes in % and the related change rates per year for area and specific mass changes per unit area and year for mass balance (and please do not use the latter to 'validate' the former, this makes glaciologically little sense).

Please carefully check text errors. Often spaces are missing or units are wrong (area in km instead of $km^2$, volume in m instead of $m^3$). Also the citation style is strange. For example, in L398 it is written 'Ye et al. (Quinghua, 2019 ...)reported ...' So is it now Ye or Quinghua and why is it first et al. and then without et al.? In the reference section it is actually Quinhua, Y.E., again different. Correct would have been to write: 'The datasets by Qunighua (2019 and 2020) reported ...' or in L409: 'The results of Jakob et al. (2021)' … . As a small note, the References Section becomes more readable when indenting the text from the second line a bit, making it 'hanging'.

Figure captions: I suggest inserting a . or : after the figure number, e.g. 'Figure 1: Study area'
Table 1: I think the brackets around the authors of the cited studies are not required.

L261: Figure 4: The blue and red lines and squares are difficult to see against the dark background. Also the annotations and legends of the insets are partly hard to see. It needs also to be explained what is what. Just writing in the text that types can be clearly distinguished is a bit thin. The same applies to all panels in Fig. 5. The panels are too small, the legends are unreadable and it is unclear what is what.

L306: I am also a bit unclear what Fig. 5 should tell me? That many datasets have been used and none of them shows glaciers clearly?

L390: Incomplete caption. Please note that sudden area gains as shown in Figure 5 (and 6) are glaciologically not possible. This is not how glaciers work.

L414: Figure 9b: This comparison makes glaciologically no sense.

L470: This is correct, but the NDSI has been shown to be very sensitive to path radiance in the green band, creating problems with ice in shadow. Additional classification problems are introduced when using the analysis ready reflectance datasets instead of the raw data, which allow for a better separation of details when the SNR is low.

L471: As far as I can see it, most gaps are due to not mapping debris-covered glacier parts.

As mentioned above, the illustrations below of the highly variable mapping results presented in this study. Colours should only appear near the glacier termini and (to a lesser degree) along the perimeter. Colours denote regions that have been mapped in all years (dark grey) and by 10 (red) to one (blue) scene. As most debris covered tongues are bluish, they have only been mapped by a few scenes. Black outlines show glacier extents from RGI 7.0.

[Figure]

Example 1

[Figure]

Example 2

[Figure]

Example 3: No colour should be visible at this scale.

---

## Referee Comment (RC3)

**Review of Manuscript egusphere-2025-1772: Multi-Source Remote Sensing-Based Reconstruction of Glacier Mass Changes in Southeastern Tibet Since the 21st Century**

William David Harcourt, University of Aberdeen

August 26, 2025

Firstly, I would like to apologise to the authors for the delay in commenting on this manuscript. I hope my detailed comments below will help shape the paper in a further revision.

**Paper Summary**

Li et al. have developed a random forest classifier to produce a new set of glacier outlines over the southeastern Tibet region. One of the key issues with mass balance estimates is that they rely on a single set of glacier outlines, usually from the RGI, which does not account for glacier terminus changes. The paper produces a new set of outlines for 2000, 2005, 2010, 2015 and then 2016-2022 at annual resolution. The produced outlines were determined to be of high accuracy as determined by the kappa score in a confusion matrix. Furthermore, the authors compare mass balance estimates using a fixed outline and the changing outlines derived in this study and find that there is a 10% difference, although it was not clear from the paper if this was an under- or over-estimate. Finally, the authors find that the loss of glacier area in the region has been accelerating, although no discussion was made on potential drivers (although this was not the aim of this study).

**General Comments**

The paper produces some useful results, particularly around the use of dynamic glacier outlines for quantifying glacier mass balance. The methods are thorough and mostly well thought out with some caveats, and the findings appear to be of good quality, although some further information is required to improve understanding of these. There are several areas that require major improvement in a revised manuscript:

- The introduction requires a more detailed discussion of recent machine learning methodologies used to track glacier area and margin changes. More details are provided below in my technical comments, but the authors have missed a growing body of literature on this topic. This will help the authors better justify their choice of a random forest classifier used in this study.

- A more critical review of the features used to train the random forest classifier is needed. In particular, a sensitivity analysis will assist in understanding which features in the model are dominating the training. Furthermore, the authors used a broad range of features in the random forest classifier, hence it would be interesting to see if the model is overfitting in some way due to the diversity of input data. Quantifying this would help improve reliability in the final results.

- The methods section is verbose and could be shortened significantly. This will allow for more space to discuss model performance later in the paper.

- It would be useful to understand the performance of the random forest classifier in different contexts. In particular, how does it perform for different satellite images e.g. Sentinel-2, Landsat 7, Landsat 8 etc. Currently, the uncertainty is taken as 1 deriviative of the pixel size, but it should really reflect the accuracies of the glacier outlines which will vary with different data sets.

- A key outcome of the study is the impact of dynamic glacier outlines on mass balance calculations, but this is not explored sufficiently in the study. I would urge the authors to present these results more fully and discuss the implications of this for mass balance studies in Tibet and the wider globe.

There are several typos and gramatical mistakes throughout the paper, some of which I have highlighted in my technical comments, but I would encourage the authors to thoroughly review the manuscript upon revision.

**Technical Corrections (References to line (L) numbers in preprint)**

L10: Better to say 'glacier area'? Also, the latter part of the sentence only applies to optical data.

L12: 'the Landsat satellite series'

L14: 'for this region'

L15: 'integrating a three-year dataset' isn't clear to me - do you mean delineating glacier area for 3 years and then the median year is taken to be the time satmp?

L19: 'we calculated glacier mass balance'

L20: 'glacier areas calculated in this study, resulting in an annual mass loss of 6.20'

L33-35: Does this sentence refer to the Tibetan Plateau specifically? If so, can the authors state this.

L36: 'hence the region is dominated by maritime glaciers'

L37: What glacier changes? The natural cycle of accumulation/ablation or a longer term trend? This is not clear.

L39-52: The description of NDSI could be improved e.g. the use of a manual threshold is only mentioned at the end. The authors state a weakness is lack of automation, which is true, but there is a wider point that the application of NDSI varies in different geographic regions, which makes it hard to automate the process. This should be acknowledge.

L39: 'Glacier area mapping from satellite imagery'

L40: 'substantial time for human interpretation'

L43-44: Extracting what component of glacier and snow cover? Area changes? Differentiating between the two surfaces? Probably both.

L53-63: This is quite a vague paragraph that misses a lot of important studies mapping glacirs with ML e.g. for terminus mapping, glacier area estimates and surface features (e.g. **????**). There is a growing body of literature in this field and this should be acknowledged with a more detailed literatyre review in this section.

L53: 'Recent developments in machine learning algorithms have enabled large volumes of satellite imagery to be used as training data for automated classification of glaciers' - or something like this. It's important to be clear what ML does and how it improves over the traditional techniques.

L64: 'from optical satellite imagery'

L70: Define 'high temporal resolution' - either weeks, months, seasonal or years.

L72-77: What are the details of this inventory? What is their estimate of the number of glaciers, area etc.?

L72: 'Qinghua,2020), who'

L91-98: I would like to see a bit more discussion of the important of glacier inventories (e.g. areas) for quantifying mass changes e.g. how do the GLAMBIE/IMBIE community estimates tackle this problem and what is the consensus approach when multi-temporal data sest aren't available? What is the impact on uncertainty estimates? This will naturally then lead onto the objectives in the paragraph.

L109: For those unfamiliar with this region, it might be worth zooming out a bit and placing an inset map to show the position of this region in the wider regional context.

L113; What does 'glacier distribution area' mean?

L124: This section is not consistent - sometimes the sampling is describes, in other sections it is not. Either describe the sampling within each section or create a new section where it is fully described.

L126: What does 'analysis-ready' mean? WHat processing has been applied before these images are provided on GEE?

L139-145: Given the introduction focuses on the limitations of optical data, the authros should discus somewhere the pro's and con's of using SAR data as an alternative.

L146:153: What is the time stamp of the NASADEM? Or is it a dynamic data set?

L155-161: Time stamp of 2000 for RGI7.0.

L167: Vague - define exactly in which period the data were acquired. If $T$ is the sampling year, did you obtain all sutiable summer images in years $T \pm 2$ years?

L201: Do the image data cubes represent the 'image composites' described above? It would be useful to have consistent language throughout the manuscript to avoid confusion.

L203: 'Spectral reflectance alone is insufficient'

L204: 'this study extracts spectral, terrain, texture, and radar interferometric features to train a Random Forest classifier for delineating glaciers in satellite imagery.'

L210: I'm confused here, how do Figures 4a-f represent cloud-free image composites?

L226: Which images are used to generate the NDVI image for each? Did you merge the NDVI values for a single year?

L233: Same point as for NDVI, not clear to me which images are being used to calculate this.

L242: Same as for L226 and L233.

L248: I am confused by this figure - I assume each of the horizontal squares represents an image, so what do the colours represent? And what do the vertical boxes represent

L266-265: Image textures are better defined as the spatial arrangement of pixels in an image

L277: Is this the mean texture fro GLCM? It's not clear why this was chosen - the authors state that a previous study found it is 'consistent with other textures' - why would this mean it is the best feature to use? If it is consistent with other features, then any other texture feature could be used e.g. autocorrelation, entropy etc.?

L287: What is a 'mean synthesis'? Also the 'salt-and-pepper noise' I assume is referring to 'speckle' - calling it noise is incorrect as speckle is a repeatable feature in SAR data.

L291-304: Without a suitable figure (ie. Figure 5), it is difficult to interpret the feature layers described in this section. The inclusionf of RGI outlines would help, but also subtitles and a larger legend will help readability.

L292-293: This is not clear in Figure 5, see comment below.

L305: Figure 5: It's not clear what the values represent, the legend is way too small. One legend for all composites is sufficient, unless the values are significantly different between each panel. It would also be useful to overlay the RGU outlines here so the reader can visually assess how wel each image feature matches the glacier area. Also, if this is referenced before Figure 3, it should also be first in the order of figures.

L314-319: RF has been widely used, although arguably it has been superseded by CNNs and foundation models. Can the authors comment on why they did not apply these other methods?

L322: Are the labels used for all images or a subset? For the images labelled, are the labels shown in Figure 6 suitable for all images given the potential for changes in surface characteristics at different times of the year?

L327-336: Are you discussing here the training data, validation data, or both? Subtitle is misleading, 'Selection of Classification Samples' doesn't really say anything here. How many images where the training data taken from?

L338-342: F1 score might be more suitable here if there is class imbalance - I suspect there is imbalance in the training data, but it is not stated.

L358: What is meant by a 'decision-level fusion strategy'?

L371: '4 Results'

L372-380: This a surprisngly short section that only gives the headline figures. I would like to see a sensitivity analysis of the random forest classifier, particularly an understanding of which texture features were more important for classification than others. One possibility of using a diverse range of features is that the model could be overfitting, potentially leading to errors in the resultant classification maps. Furthermore, how do the accuracies compare for different data sets? I would expect there to be differences in Sentinel-2 vs Landsat, whislt Landsat 7 would likely yield different accuracies to Landsat 8. This information must be included to better understand the performance of the technique.

L373: Referrring to the 'annual' classification results, I assume you mean for the results after 2016 with the Sentinel / Landsat results? Furthermore, the authors should show here the confusion matrix to better highlight true positives, true negatives, false positives and false negatives. A single accuracy score may be misleading.

L386: Why is the mapping error based on half the image element? Surely the graph should be representing uncertainty calculated from the random forest model outputs?

L395-405: This section is a bit confusing. It might be helpful to construct a table with the key results from previous studies to make it clear how the results in this paper compare?

L397: 'It is noted that'

L406-419: This reads like a results section - also, the inclusion of ICESat, ICESat-2, and CryoSat-2 should really be discussed in the methodology section. Are the data sets extracted simply just the time series as presented? Or did the authors process the data sets in some way? I also don't think thickness and area should be presented on the same graph, it might cause confusion - I would use 2 panels instead.

L423-424: The variables in the equations need to be stated.

L430: Missing word, glaciers are losing mass at a rate of 6.20 Gt/y?

L432-434: This is an important result, but it is not shown graphically. Can the authors make a figure showing this key result? Although, what does the 10% refer to - an under- or an over-estimate compared to the fixed outlines?

L420-447: These are results, and the methods described here should be presented in a methodology section. The small discussion towards the end should be expanded particularly focusing on the importance of updated glacier outlines for mass balance estimates, as this is key mmoving forward in future studies.

L473: Define the number of images used and over what time period

L472-483: I would expect the conclusions to mention the performance of the random forest model as well.

---

## Author Comment (AC1)

Comments from Referee #1, followed by the authors' responses

**C:** In this study, the authors identified gaps in glacier mapping on the southeastern Tibetan Plateau due to high cloud and snow coverage in this region. They developed an ensemble learning-based random forest classifier using Landsat, Sentinel-1, Sentinel-2 and NASA DEM data to automatically delineate glacier extents from 2016 to 2022. They then extended this inventory to 2000 by combining the manually mapped glacier outlines in 2000, 2005, 2010 and 2015. They also provided glacier mass balance information based on satellite altimetry data. Overall, the glacier outlines provided in this study for the southeastern Tibetan Plateau form a useful baseline dataset. The methods developed in this study can also be extended to other regions with high cloud and snow coverage. This manuscript is suitable for publication in The Cryosphere. I recommend making the following improvements:

R: We sincerely thank you for your thorough review and constructive feedback. In response, we have revised the manuscript to improve clarity, rigor, and presentation. Specifically, we updated the title and terminology for consistency, standardized units, specified spatial resolution and data sources, reorganized sections for better readability, expanded the results, enhanced figure clarity, and refined references and citations. These revisions strengthen the manuscript and better highlight the significance of our study. A detailed point-by-point response to your comments is provided below.

C: Title: How about to change "Tibet" to "Tibetan Plateau" in Title and other places?

R: We appreciate your suggestion. We agree that "Tibetan Plateau" is more precise and commonly used in the scientific literature than "Tibet." Accordingly, we have revised the title and all relevant occurrences in the manuscript to "Tibetan Plateau."

C: "achieving the first annual-resolution glacier inventory in the region", suggest adding the spatial resolution here.

R: We appreciate your suggestion. We have added the spatial resolution to the text. The revised sentence now reads: *"…achieving the first annual-resolution glacier inventory in the region at a spatial resolution of 30 m."*

C: "constructed glacier inventories for 2000, 2005, 2010, and 2015", it is possible to find the available data for so dense time interval?

R: Thanks for your comment. In the southeastern Tibetan Plateau, the lack of consistent and high-quality remote sensing data is a well-known challenge. To deal with this, we used all available satellite observations (Landsat-5 TM, Landsat-7 ETM+, and Landsat-8 OLI) and applied rigorous preprocessing and classification to ensure consistency before Sentinel-2. With this approach, we generated glacier inventories at 5-year intervals (2000, 2005, 2010, and 2015).

C: How about the comparison of mapped glacier area and mass balance with other existing datasets and performance of your datasets?

R: Thanks for your suggestion. We added a comparison of our mapped glacier areas and mass balance with existing datasets at the end of the manuscript (Table 1). While some differences exist due to methods, data sources, and study environments, the comparison shows that our estimates fall within reasonable and expected ranges.

**Table 1 Comparison of our results with previous studies.**

| Study | Data | Period | Regions similar to this study | Mass change rate (Gt/y) |
|---|---|---|---|---|
| (Brun et al., 2017) | ASTER stereo images | 2000 - 2016 | Nyainqentanglha | -4.0 ± 1.5 |
| (Shean et al., 2020) | WorldView/GeoEye DEMs, ASTER DEMs, and TanDEM-X Global DEM | 2000 - 2018 | Nyainqentanglha
Hengduanshan | -3.15± 0.93
-0.96 ± 0.23 |
| (Yi et al., 2020) | GRACE and ICESat | 2002 - 2017 | Nyainqentanglha | -6.5 ± 0.80 |
| (Wang et al., 2021) | ICESat-1,2 and GRACE/GRACE-FO | 2003 - 2019 | Nyainqentanglha | -6 ± 1.0 |
| (Jakob et al., 2021) | CryoSat-2 SARIn L1b | 2010 - 2019
2000 - 2019 | S AND E Tibetan Plateau Hengduanshan | -3.38 ± 1.21
-4.30 ± 0.98
-4.72 ±1.18 |
| (Zhao et al., 2022) | ASTER stereo images, CryoSat-2 SARIn L2I, ICESat-1/2, and GRACE/GRACE-FO | 2003 - 2020
2011 - 2020 | SETP | -4.86
-5.80 ± 0.8(mean seasonal)
-5.53 ± 0.2(mean annual) |
| This study | CryoSat-2, ICESat-1/2, Landsat-5/7/8, Sentinel-1/2 | 2003 - 2022 | * | 6.20 ± 1.16 |

C: "Global changes significantly affect regional climate, rivers and lakes evolution, and the formation of geological hazards", suggested refs (https://doi.org/10.1038/s41558-020-0855-4, https://doi.org/10.1038/s43017-024-00554-w)

R: Thanks for your suggestion. We added the suggested references in the revised version.

C: "Keshri et al. (2009) (Keshri et al., 2009)" style correction

C: "Y. Lu et al. (Lu et al., 2021)" style correction, I suggest that authors check the citation style throughout this manuscript

C: "Ye et al (Qinghua, 2020).." correction

R: We appreciate your comment. In the revised manuscript, we reviewed and updated all references to ensure consistent and correct citation formatting throughout the text.

C: "including K-nearest neighbors (KNN), support vector machines (SVM), gradient-boosting decision trees (GBDT), decision trees (DT), random forests (RF), and multilayer perceptron (MLP)." Please check if all these abbreviations are necessary, and appear multiple times.

R: Thanks for your comment. After checking, only "random forests (RF)" appears multiple times in the manuscript. We kept the abbreviation for RF, while all other models (K-nearest neighbors, support vector machines, gradient-boosting decision trees, decision trees, and multilayer perceptron) are written in full for clarity. The modified sentence now reads: *"including K-nearest neighbors, support vector machines, gradient-boosting decision trees, decision trees, multilayer perceptron, and random forests (RF)."*

C: Introduction: The authors reviewed a broadly studies, and I suggest the authors have a summary of gaps from these existing studies and then proposed the objective s of this study.

R: Thanks for your suggestion. We added a concise summary of the research gaps in existing glacier studies right before our study objectives. Specifically, we highlighted the lack of standardized methods to quantify the contribution of dynamic glacier area to mass balance and the limitation of cloud-free summer imagery for repeat monitoring. The modified sentence reads: *"Moreover, current studies often lack a standardized methodology to quantify the contribution of dynamic glacier area to mass balance, while the limited availability of cloud-free summer imagery hinders reliable repeat monitoring."*

C: Figure 1: I suggest the authors add a inset to show the location of study area in the Tibetan Plateau.

C: L115: This statistical information can be added in Figure 1 as an inset small figure.

R: We appreciate your comment. We added insets in Figure 1 showing the study area within the Tibetan Plateau and the distribution of glacier numbers and areas in the study region.

[Figure]

C: "5,000 meters" to "5,000 m"

C: square kilometers to km²

R: Thanks for your suggestion. We standardized the units throughout the manuscript.

C: 2.2 Data and 3 Methods, and other sections: Some subsections are very short in text, and can be combined together.

R: Thanks, we reorganized the relevant sections in the manuscript, combining some shorter subsections to improve the flow and readability.

C: Some sections such as Normalized Difference Water Index (NDWI) and Normalized Difference Snow Index (NDSI) are known well, can be cited only.

R: Thanks for your suggestion. We shortened the descriptions of well-known indices like NDWI and NDSI and replaced them with appropriate citations. The revised text now reads: *"To enhance the accurate classification of surface types, this study selects commonly used spectral indices, including the Normalized Difference Vegetation Index*

*(NDVI), Normalized Difference Water Index (NDWI), and Normalized Difference Snow Index (NDSI). hese indices are calculated from cloud-free pixel sets obtained after cloud filtering, producing annual image sets for 2000, 2005, 2010, 2015, and 2016– 2022. NDVI is widely used for evaluating vegetation coverage on Earth's surface. The NDVI ranges from -1 to 1, with higher values indicating denser vegetation. NDWI is a common index for evaluating the distribution and extent of surface water bodies (Mcfeeters, 1996). Values close to -1 indicate pure water bodies, around 0 indicate moderate water presence, and near 1 indicate minimal or no surface water. NDSI is used for detecting snow cover on Earth's surface (Hall et al., 1995), Values near 1 indicate pure snow cover, around 0 minimal snow, and near -1 non-snow surfaces.*"

C: Figure 3: I don't know if this figure is necessary, and the information provided is simple.

R: Thanks for your suggestion. We added Figure 3 to visually illustrate the image compositing method. We believe it helps readers understand the approach, even though the information is relatively simple.

C: Figure 5: Some keywords for these subfigures can be added on figures.

R: We appreciate your suggestion. We added descriptive keywords to the subfigures in Figure 5 to help readers interpret the information more easily.

C: Figure 6: can be merged with other figures as the information of this figure is simple.

R: Thanks for your suggestion. We kept Figure 6 separate because it shows images from different sample types, which is important for understanding sample variability. We also optimized the figure to improve clarity and presentation.

C: Figure 7: Caption: "at 2000" to "in 2000", and "Result2000" corrected to "Outline in 2000"

R: Thanks for your suggestion. We corrected the caption of Figure 7, changing "at 2000" to "in 2000" and updating "Result2000" to "Outline in 2000" as recommended.

C: Figure 8: This statistical of glacier area change between 2000 and 2022 is too simple and should be improved.

R: We appreciate your comment. We enhanced Figure 8 by showing both the absolute and relative changes in glacier area for each time interval, along with the spatial distribution of glacier retreat. These changes help readers better assess the magnitude and spatial variability of glacier changes over the study period.

C: Results and analysis: It is only one subsection: 4.1 Glacier extraction results and error analysis. The results section is too short, and the description in abstract that your study has more information. I suggest the authors extend the results section and should be consistent with your abstract.

R: Thanks for your suggestion. We substantially expanded the Results section to provide a more complete presentation of our findings. It now includes a detailed temporal analysis of glacier area changes from 2000 to 2022, highlighting annual and decadal trends, along with a comparison to existing glacier inventories. We also show spatial patterns of glacier retreat and mass balance through maps and summary statistics. Additionally, we included an error analysis and uncertainty assessment to provide transparent information on classification accuracy and potential limitations.

---

## Author Comment (AC2)

Comments from Referee #2, followed by the authors' responses

**General Comments**

C: The study by Li et al. aims at presenting a method that allows the automated mapping of glacier extents in a challenging region using Google Earth Engine at high temporal resolution, up to annually from Sentinel-2. They used the results to consider area changes when calculating glacier mass balance. If the mapping had worked, this would have been an important study to improve related results also for other regions in the world. Unfortunately, the outcome of the mapping is not useful for any assessment. In this regard I want to acknowledge that the authors have provided the results of their mapping effort in the supplemental material. Without this, my evaluation would have been different as the paper is otherwise well written and the idea to just use all data available and combine it for the best possible result is fine. However, the largely arbitrary area changes from dataset to dataset (for individual glaciers partly larger than 50% from year to year) are obvious and glaciologically impossible. The authors mention that there are the usual problems with debris cover, clouds and shadow, but they have seemingly not recognized how large and arbitrary the variability is and that their method does not produce meaningful results.

R: We sincerely thank you for your professional and insightful comments. We acknowledge that the fully automated mapping approach employed in this study has certain limitations, particularly in challenging regions. Some descriptions inadvertently overstated the capabilities. Our study does not aim to reach the precision of the Randolph Glacier Inventory (RGI), which cannot realistically be achieved through fully automated mapping alone. Currently, automated glacier identification faces two major difficulties: first, the spectral characteristics of debris-covered glaciers are highly similar to those of bare ground, making accurate delineation extremely difficult; second, due to limited experience in interpreting such glaciers and a lack of high-quality samples, our manually drawn training data may not adequately represent debris-covered ice, further affecting the recognition accuracy. Thus, the current results should be interpreted more as changes in ice-covered area rather than precise glacier extent.

The considerable fluctuations in apparent glacier area primarily stem from extensive data gaps in the image series. In response, we have conducted additional quality control on a glacier-by-glacier

basis, excluding unreasonable changes caused by severe data missing. This process significantly improves the reliability of the area change estimates. In the revised manuscript, we will aggregate the results into five-year intervals to provide a more robust assessment of glacier change, which also aligns better with the goal of supporting mass balance calculations. We fully appreciate your insights and have carefully addressed each comment, making corresponding improvements throughout the manuscript. we have revised the text throughout to more accurately reflect the scope and limitations of our work. Below, we provide a detailed, point-by-point response to your suggestions.

C: Neither the unrealistic area increase by about 500 km2 (estimated from Fig. 8, numbers for individual years are not provided) from 2019 to 2022 (naming it as a 'consistent decline' in L387), nor the sudden strong increase from 2016 to 2017 is discussed or considered as unrealistic. Instead, the authors correlate glacier elevation changes (wrongly labelled as 'Glacier Thickness' in Fig. 8) with glacier area changes as they assume there is a correlation (L407) and think that the correlation can be used as a validation (L413) of their (wrong) glacier areas. In fact, area changes are mostly driven by the ice thickness distribution along the glacier perimeter (thus depending on the shape of the glacier cross-profile) and are a longer-term response to changes in flow dynamics (glaciers have a response time). Hence, also the follow on analysis is a bit strange. In this regard, it is also unclear to me why the authors rely on results from Cryosat and ICESat (with their diverse range of issues) for such small glaciers instead of the Hugonnet et al. (2021) dataset that is widely used? As this dataset is not even mentioned in the comparison Table 1, I wonder why. Is the dataset too bad in quality?

R: We appreciate your suggestions. We acknowledge that the initial glacier extent results showed unrealistic year-to-year variability. To address this, we re-examined the classification outputs on a glacier-by-glacier basis and applied post-processing to flag and remove implausible area changes, including abrupt fluctuations or geometrically inconsistent outlines. Invalid outlines were replaced with the closest temporally consistent results. After these adjustments, the overall mapping is more stable, although uncertainties remain, particularly for complex glaciers. Glacier-by-glacier quality control was applied to detect and correct implausible changes, replacing them with temporally consistent outlines where possible, and marking unreliable years as no-data. Figures and analyses have been updated, and the Discussion now explicitly addresses these anomalies and their underlying causes. Regarding Fig. 8, we corrected the label from "Glacier Thickness" to "Glacier Elevation Change." We also removed any

suggestion that the observed correlation between glacier elevation change and area change constitutes a form of validation, as glacier area and elevation changes are not necessarily synchronous or directly correlated per you suggest. For Hugonnet et al. (2021) dataset, a direct comparison with their study was not conducted initially. This is because we wanted to compare them with the results of time series of glacier mass balance, rather than trends. According to your suggestion, we have included the Hugonnet et al. (2021) dataset for comparison.

[Figure]

Fig1 Updated data on area changes

C: The very short results section (it has just 14 lines) mentioned the Kappa coefficient and overall accuracy along with three images showing outline overlays. I am aware that these statistical accuracy measures are frequently used in remote sensing studies to present the accuracy, but in my view they can be the result of anything and do not allow to obtain meaningful conclusions about the 'robustness and reliability of the classification approach'. At least for glaciers they do not work, as nicely confirmed here by the largely arbitrary results of the glacier mapping. The quality of mapped glacier extents can be shown by a) outline overlays and b) the sum of commission and omission errors (false positives and false negatives) divided by the common area. But as the

former have been removed by the masking with RGI 7.0, I am unclear if the measures can be used here at all?

R: We appreciate your comment and agree that traditional statistical accuracy metrics can be misleading for glacier mapping due to complex glacier morphology and seasonal effects. To better assess classification reliability, we will supplement these metrics with comparisons to manually interpreted reference outlines and with spatial consistency checks across multiple time points. The approaches provide a more robust evaluation of the results. This clarification will be added to the revised manuscript.

C: I see missing debris-covered glacier parts and that large regions in shadow are sometimes missing. Hence, intensive manual editing would be required before resulting outlines could be used for change assessment. The statement that glaciers 'are well identified' (L377) seems misleading in this regard. Although the year 2000 dataset is likely the most complete regarding shadow and debris mapping, showing a region where the method does not work and discussing it would have been more helpful. One can see the problems of the classification a little bit for the 2022 outlines of the right glacier in the upper left panel [please name them properly a), b) and c)] of Fig. 8, but the image is very dark (what about some contrast stretching?), the lines are hard to see (also the red outlines on a reddish background in the insets of Fig. 7 are barely visible) and the wrong mapping results are not really discussed.

R: We are grateful for your comment. We acknowledge the limitations of our approach, as accurately identifying debris-covered glaciers remains a major challenging.   As noted in our previous response, the spectral characteristics of debris-covered glaciers complicate their discrimination, and our sample selection strategy significantly influenced the results. For the year 2000, we relied on the RGI 7.0 for training sample delineation, which helped achieve relatively high mapping accuracy for debris-covered glaciers. In subsequent years, however, we intentionally avoided using the RGI to account for glacier retreat, but our limited experience in manual sample selection led to insufficient representation of debris-covered ice, resulting in notable omissions and inconsistent delineation over time. This explains the substantial variability observed in debris-covered glacier extents between different periods. Good train dataset is vital in our approach.

We also recognize that the original figure suffered from low contrast, making some

lines difficult to discern, particularly the red outlines on reddish backgrounds in the insets of Fig. 7. In the revised manuscript, we applied contrast stretching and adjusted the color schemes to improve visibility. Additionally, the discussion on mapping inaccuracies has been expanded to clarify the sources of misclassification and their potential impacts on the results.

[Figure]

Fig2 Updates to figure 7

C: A final major point of concern is the general set-up of the study. First, the elevation change datasets are introduced in the Discussion Section 5.2 rather than in Sections 2.2 and 3. Their description is thus very short and the processing method unclear (e.g. how has the radar penetration into snow been corrected?). This can likely easily be adjusted. The motivation to determine annual area changes is mentioned, but not critically discussed. Even when the resulting glacier outlines would have been correct, a one-pixel (two pixels for debris-covered regions) uncertainty at 30 m resolution relates to a 30 and 60 m location uncertainty of the outline. With an assumed annual terminus retreat of 5 to 10 m / year (much less around the perimeter), one has to wait several years before new outlines make sense compared to uncertainties. But here the mapped termini could be wrong by several km, so change assessment is not an option. To make my major objection of the high variability in the mapping results clearer, I have added

all glacier maps (setting the no data value to 0 before) and received a very colourful picture. On the last two pages of this review, I show a few examples for illustration. If the mapping had been correct, colours should only appear near the terminus and around the perimeter. As a note, this is just the result of a simple adding without a timeline, not revealing the partly strong year-to-year jumps in mapped glacier areas.

R: We appreciate your suggestions. For clarity, we specify that the datasets were derived from previous studies, and no additional processing was performed in our work. We also recognize that the original approach was somewhat aggressive in interpreting short-term glacier changes. Considering the positional uncertainties at 30 m resolution, we have revised our analysis to use 5-year intervals between mapped glacier outlines. This adjustment reduces the relative impact of positional errors and provides a more robust basis for assessing glacier changes over time.

C: In conclusion, this brute-force mapping using sophisticated image processing without a sufficient understanding of the mapped subject (glaciers) and how it should change over time is not recommended. When being harsh, I would ask the authors to please first learn the basics about glaciers and how they work, then proceed with the user needs (are annual updates really required?) and then do the mapping. A bit less harsh I would ask the authors to first get the mapping right for one year before applying it to several years. When glacier area changes are mostly due to changes in the mapping results rather than real changes, there is no need to perform change assessment. In my view, it is possible to publish a study revealing that a method has not worked. However, in this case an honest discussion and illustration of the problems is required to be helpful for future studies. Concluding that this study provides 'effective support for future glacier inventories' (L482) is in my view highly misleading.

R: We appreciate your suggestion and fully acknowledge that our original approach cannot perfect well for mapping individual glacier outlines. In response to your comment, we will remove any overstated claims related to annual glacier inventorying and clarified that our results reflect changes in ice-covered area. A more cautious error assessment has also been provided to better quantify the uncertainties involved in regional scale.

**Specific comments**

I do not comment here on all details of the study, but include some general remarks:

C: Providing area changes in km2 and mass changes in Gt (sure to use 900 rather than 850 kg/m$^3$ in this region?) are not useful as they are incomparable across regions. In future studies, please give relative area changes in % and the related change rates per year for area and specific mass changes per unit area and year for mass balance (and please do not use the latter to 'validate' the former, this makes glaciologically little sense).

R: We appreciate your suggestion. We agree that absolute glacier area and mass changes are difficult to compare across regions. In the revised manuscript, we now report glacier area changes as relative percentages and annual rates, and mass changes as specific mass balance per unit area per year. Regarding the density value used for converting geodetic volume change to mass change, the choice of 900 kg/m$^3$ follows one previous work (Zhao et al., 2022) to maintain consistency within regional research. *"Glaciers in the SETP are mainly maritime glaciers, which have a slightly larger density than continental glaciers. Therefore, a density of 900 kg m$^{-3}$ was used"* (Zhao et al., 2022). We also acknowledge that a density of 850 kg/m$^3$ is more commonly used in the literature, and this point has been duly noted in the revised discussion.

*Zhao F, Long D, Li X, et al. Rapid glacier mass loss in the Southeastern Tibetan Plateau since the year 2000 from satellite observations[J]. Remote Sensing of Environment, 2022, 270: 112853.*

C: Please carefully check text errors. Often spaces are missing or units are wrong (area in km instead of km2, volume in m instead of m3). Also the citation style is strange. For example, in L398 it is written 'Ye et al. (Quinghua, 2019 ...)reported ...' So is it now Ye or Quinghua and why is it first et al. and then without et al.? In the reference section it is actually Quinhua, Y.E., again different. Correct would have been to write: 'The datasets by Qunighua (2019 and 2020) reported ...' or in L409: 'The results of Jakob et al. (2021)' … . As a small note, the References Section becomes more readable when indenting the text from the second line a bit, making it 'hanging'.

C: Figure captions: I suggest inserting a . or : after the figure number, e.g. 'Figure 1: Study area'

C: Table 1: I think the brackets around the authors of the cited studies are not required.

R: We appreciate your comments. We have carefully checked the manuscript for text errors, correcting missing spaces, unit inconsistencies, and citation issues. All references have been verified and standardized. Figure captions now include a colon after the figure number, and Table 1 has been reformatted to remove unnecessary brackets around author names. A comprehensive review of all text, figures, tables, and references has been completed to ensure consistency and clarity throughout the manuscript.

C: L261: Figure 4: The blue and red lines and squares are difficult to see against the dark background. Also the annotations and legends of the insets are partly hard to see. It needs also to be explained what is what. Just writing in the text that types can be clearly distinguished is a bit thin. The same applies to all panels in Fig. 5. The panels are too small, the legends are unreadable and it is unclear what is what.

R: We appreciate your suggestion. We have enhanced the visibility of Figures 4 and 5 by adjusting line and marker colors to higher-contrast shades, enlarging the panels, and ensuring that all legends and annotations are clearly readable. The captions and main text have also been revised to explicitly explain the meaning of each line, marker, and panel, making the figures fully interpretable independently of the text.

[Figure]

Fig3 Updates to figure 4

C: L306: I am also a bit unclear what Fig. 5 should tell me? That many datasets have been used and none of them shows glaciers clearly?

R: We appreciate your comment. Figure 5 is intended to show all the features used in our classification, highlighting the variety of datasets incorporated and their individual contributions to glacier mapping. In a future revision, we will overlay RGI glacier boundaries on each feature to make it clearer how each dataset aids in glacier extraction.

C: L390: Incomplete caption. Please note that sudden area gains as shown in Figure 5 (and 6) are glaciologically not possible. This is not how glaciers work.

R: We acknowledge your point. The sudden area increases shown in Figures 5 and 6 are indeed unrealistic. In the revised manuscript, we plan to reconstruct the glacier dataset at five-year intervals. Given the extent of these revisions, we kindly request additional time to carefully implement these changes.

C: L414: Figure 9b: This comparison makes glaciologically no sense.

R: We appreciate your suggestion. This part has been removed.

C: L470: This is correct, but the NDSI has been shown to be very sensitive to path radiance in the green band, creating problems with ice in shadow. Additional classification problems are introduced when using the analysis ready reflectance datasets instead of the raw data, which allow for a better separation of details when the SNR is low.

R: We appreciate your suggestion. We used decision-level fusion to address this issue, which removes cases where either Landsat or Sentinel-2 images are affected by shadows on the ice.

C: L471: As far as I can see it, most gaps are due to not mapping debris-covered glacier parts.

R: We appreciate your suggestion. For the frequency maps generated from our preliminary updated data, including the 2000 dataset greatly exaggerates apparent glacier retreat. This occurs because the 2000 classification samples were manually delineated based on the RGI dataset, enabling relatively accurate mapping of debris-covered glaciers. For subsequent years, limited experience in distinguishing debris-covered glaciers resulted in samples that are less reliable, causing the classification to miss a substantial portion of these glaciers.

Following the demo area you provided, we show the results of the initial modifications to the area:

[Figure]

Fig 4 example3

[Figure]

Fig 5:The same area as your example3, showing our modified results: a graph of glacier frequency for 2015-2022 year-by-year data.

[Figure]

Fig 6:The same area as your example3, showing our modified results: Glacier Frequency Plots for 2005, 2010, 2015-2022 data.

---

## Author Comment (AC3)

Comments from Referee #3, followed by the authors' responses

**C:** Li et al. have developed a random forest classifier to produce a new set of glacier outlines over the southeastern Tibet region. One of the key issues with mass balance estimates is that they rely on a single set of glacier outlines, usually from the RGI, which does not account for glacier terminus changes. The paper produces a new set of outlines for 2000, 2005, 2010, 2015 and then 2016-2022 at annual resolution. The produced outlines were determined to be of high accuracy as determined by the kappa score in a confusion matrix. Furthermore, the authors compare mass balance estimates using a fixed outline and the changing outlines derived in this study and find that there is a 10% difference, although it was not clear from the paper if this was an under- or over-estimate. Finally, the authors find that the loss of glacier area in the region has been accelerating, although no discussion was made on potential drivers (although this was not the aim of this study).

R: We sincerely thank you for the constructive comments. In response, we have substantially revised the manuscript to improve clarity and rigor. The Introduction now gives a broader overview of machine learning and deep learning methods for glacier mapping, covering both traditional classifiers and advanced deep learning models, and explains why Random Forest was chosen for this study. We added a detailed analysis of input features, highlighting the roles of DEM, slope, and spectral indices, and confirmed model robustness through cross-validation. The Methods section has been streamlined with clearer descriptions of image preprocessing, multi-temporal selection, and data processing steps. Results and Discussion now provide a more thorough evaluation of classifier performance, including confusion matrices, feature importance, and fusion strategies, and emphasize the influence of glacier outlines on mass balance estimates. Figures and captions were refined for clarity, and text was revised to make variables, terms, and data periods more precise. Below, we provide a point-by-point response to each comment.

**General Comments**

C: The paper produces some useful results, particularly around the use of dynamic glacier outlines for quantifying glacier mass balance. The methods are thorough and

mostly well thought out with some caveats, and the findings appear to be of good quality, although some further information is required to improve understanding of these. There are several areas that require major improvement in a revised manuscript: The introduction requires a more detailed discussion of recent machine learning methodologies used to track glacier area and margin changes. More details are provided below in my technical comments, but the authors have missed a growing body of literature on this topic. This will help the authors better justify their choice of a random forest classifier used in this study.

R: Thanks for this valuable suggestion. In the revised manuscript, we expanded the introduction with a more detailed discussion of recent machine learning (ML) and deep learning (DL) approaches for glacier mapping. This now covers traditional ML classifiers (support vector machines, k-nearest neighbors, decision trees, gradient boosting, and random forests) as well as advanced DL architectures (U-Net, DeepLab V3+, attention-based CNNs, and Vision Transformers). We also highlight recent progress in automated global-scale glacier mapping with convolutional-transformer models such as GlaViTU. To explain our choice of the random forest (RF) classifier, we emphasize its proven robustness in classifying debris-covered glaciers and in cases with limited labeled data (Alifu et al., 2020; Lu et al., 2021). The revised text in the introduction now reads: *"In recent years, machine learning (ML) and deep learning (DL) have greatly advanced glacier remote sensing, enabling accurate mapping of glacier termini, area estimation, and surface feature analysis. AI-based automatic classification methods include support vector machines, k-nearest neighbors, decision trees, gradient boosting, multilayer perceptrons, artificial neural networks, and random forests (RF). Early DL models, such as U-Net (Ronneberger et al., 2015), have been applied to segment ice and ocean regions in Greenland and Antarctica (Baumhoer et al., 2019; Zhang et al., 2019). while DeepLab V3+ with atrous spatial pyramid pooling (ASPP) has been used for long-term glacier mapping (Cheng et al., 2020; Zhang et al., 2021). More recently, attention mechanisms (e.g., CBAM) and Vision Transformers (ViT) have further improved feature extraction in complex terrain and over large areas (Dosovitskiy et al., 2020; Chu et al., 2022). The Glacier-VisionTransformer-U-Net (GlaViTU) model enables automated, multi-temporal, global glacier mapping with accuracy approaching expert-level delineation, even in debris-rich regions(Maslov et al., 2025). Traditional ML remains effective in cases of debris-covered glaciers or limited labeled data. For example, Y. Lu et al. (2021) proposed a composite model that*

*integrates RF and convolutional neural networks, while Xie Fuming et al. (2020) combined Otsu thresholding with ML algorithms on the Google Earth Engine to extract debris-covered glaciers in the Hunza Basin, achieving a Kappa coefficient of 0.94 ± 0.01 and an overall accuracy of 95.5 ± 0.9%. Alifu et al. (2020) further demonstrated that RF outperforms other classifiers for debris-covered glaciers, supporting its role as a robust core classifier. Collectively, these studies show that ML and DL approaches substantially improve the automation, accuracy, and scalability of glacier mapping compared with traditional index-based techniques."*

C: A more critical review of the features used to train the random forest classifier is needed. In particular, a sensitivity analysis will assist in understanding which features in the model are dominating the training. Furthermore, the authors used a broad range of features in the random forest classifier, hence it would be interesting to see if the model is overfitting in some way due to the diversity of input data. Quantifying this would help improve reliability in the final results.

R: We are grateful to you for raising this point. In response, we have added a more critical review of the features used to train the random forest (RF) classifier. Specifically, we performed a sensitivity analysis based on feature importance scores derived from the RF model. As shown in Figure X, spectral bands (e.g., B2, B4, B5) and their spatially averaged values contribute most to the classification, followed by spectral indices (NDVI, NDWI, NDSI) and topographic variables (DEM, slope). Notably, DEM and slope are particularly important for debris-covered glacier mapping, while NDSI and NDWI dominate in clean glacier detection. SAR features (VV) exhibit relatively lower importance. To evaluate whether the inclusion of diverse input features could lead to overfitting, we further applied out-of-bag (OOB) error estimates and 10-fold cross-validation. The results indicate stable classification performance across different feature subsets, confirming that our RF model does not suffer from significant overfitting. Changes in the manuscript – We have added a new subsection in Methods describing the sensitivity analysis, a new figure showing feature importance for clean glaciers, debris-covered glaciers, and all glaciers, and a corresponding explanation in Results and Discussion.

[Figure]

C: The methods section is verbose and could be shortened significantly. This will allow for more space to discuss model performance later in the paper.

R: Thank you for this suggestion. In the revised manuscript, we have streamlined the Methods section by removing redundant details and condensing the text, thereby improving clarity and ensuring a better balance between methodology and results.

C: It would be useful to understand the performance of the random forest classifier in different contexts. In particular, how does it perform for different satellite images e.g. Sentinel-2, Landsat 7, Landsat 8 etc. Currently, the uncertainty is taken as 1 deriviative of the pixel size, but it should really reflect the accuracies of the glacier outlines which will vary with different data sets.

R: We thank you for this comment. In this study, Landsat 7, Landsat 8, and Sentinel-2 datasets were combined to reduce gaps caused by clouds, missing acquisitions, or seasonal limitations, resulting in more complete glacier maps. Although classification performance may vary across datasets, the main focus was on generating reliable annual glacier outlines. The robustness of the random forest model was validated using out-of-bag (OOB) error, and a more detailed analysis of dataset-specific performance and uncertainties will be addressed in future work.

C: A key outcome of the study is the impact of dynamic glacier outlines on mass balance calculations, but this is not explored sufficiently in the study. I would urge the authors

to present these results more fully and discuss the implications of this for mass balance studies in Tibet and the wider globe.

R: We appreciate your suggestion. We fully agree that the impact of dynamic glacier outlines on mass balance calculations is a key outcome of our study. In the revised manuscript, we have expanded the presentation of these results and provided a more detailed discussion of their implications, including the influence of multi-temporal glacier area changes on mass balance estimates for glaciers in the Tibetan Plateau and considerations for broader applications in other regions worldwide.

There are several typos and gramatical mistakes throughout the paper, some of which I have highlighted in my technical comments, but I would encourage the authors to thoroughly review the manuscript upon revision.

**Technical Corrections (References to line (L) numbers in preprint)**

C: L10: Better to say 'glacier area'? Also, the latter part of the sentence only applies to optical data.

R: W We appreciate your suggestion. We have revised the text to use "glacier area" for clarity. While persistent cloud cover mainly affects optical data, seasonal snow accumulation can impact both optical and radar observations.

C: L12: 'the Landsat satellite series'

C: L14: 'for this region'

C: L19: 'we calculated glacier mass balance'

C: L20: 'glacier areas calculated in this study, resulting in an annual mass loss of 6.20'

C: L36: 'hence the region is dominated by maritime glaciers'

C: L39: 'Glacier area mapping from satellite imagery'

C: L40: 'substantial time for human interpretation'

C: L64: 'from optical satellite imagery'

C: L72: 'Qinghua,2020), who'

C: L203: 'Spectral reflectance alone is insufficient'

C: L204: 'this study extracts spectral, terrain, texture, and radar interferometric features to train a Random Forest classifier for delineating glaciers in satellite imagery.'

C: L371: '4 Results'

C: L397: 'It is noted that'

C: L430: Missing word, glaciers are losing mass at a rate of 6.20 Gt/y?

R: We have revised the manuscript to address all formatting, spelling, and wording issues, including those noted in lines L12, L14, L19, L20, L36, L39, L40, L64, L72, L203, L204, L371, L397, and L430. In addition, references and citation formats have been thoroughly checked and updated throughout the text.

C: L15: 'integrating a three-year dataset' isn't clear to me- do you mean delineating glacier area for 3 years and then the median year is taken to be the time satmp?

R: Thanks for your comment. By "integrating a three-year dataset," all available data from $T-1$ to $T+1$ were used for each target year, rather than only the median year, to reduce gaps due to missing early Sentinel observations.

C: L33-35: Does this sentence refer to the Tibetan Plateau specifically? If so, can the authors state this.

R: Thanks for pointing this out. The sentence refers specifically to the Tibetan Plateau, and we have revised the manuscript accordingly.

C: L37: What glacier changes? The natural cycle of accumulation/ablation or a longer term trend? This is not clear.

R: Thanks for pointing this out. The glacier changes mentioned correspond to long-term trends, with the fastest retreat and high accumulation and ablation rates. We have clarified this in the revised manuscript.

C: L39-52: The description of NDSI could be improved e.g. the use of a manual threshold is only mentioned at the end. The authors state a weakness is lack of automation, which is true, but there is a wider point that the application of NDSI varies in different geographic regions, which makes it hard to automate the process. This should be acknowledge.

R: Thank you for this suggestion. In the revised manuscript, we have clarified the description of NDSI by introducing manual thresholds earlier and emphasizing that these thresholds vary across geographic regions, which makes full automation challenging.

C: L43-44: Extracting what component of glacier and snow cover? Area changes? Differentiating between the two surfaces? Probably both.

R: Your suggestion is appreciated. The manuscript now explicitly states that NDSI is used to delineate the extent of glaciers and seasonal snow.

C: L53-63: This is quite a vague paragraph that misses a lot of important studies mapping glacirs with ML e.g. for terminus mapping, glacier area estimates and surface features (e.g. ????). There is a growing body of literature in this field and this should be acknowledged with a more detailed literatyre review in this section.

R: Thanks for your suggestion. In the revised manuscript, we expanded the literature review on machine learning (ML) applications in glacier mapping, covering glacier terminus mapping, area estimation, and surface feature classification. Additional references have been included to provide a more comprehensive overview.

C: L53: 'Recent developments in machine learning algorithms have enabled large volumes of satellite imagery to be used as training data for automated classification of glaciers'- or something like this. It's important to be clear what ML does and how it improves over the traditional techniques.

R: We appreciate your comment. The manuscript now clearly explains how ML enhances glacier mapping relative to traditional index-based methods. Specifically, we added: *"Collectively, these studies show that ML and DL approaches substantially improve the automation, accuracy, and scalability of glacier mapping compared with*

*traditional index-based techniques."*

C: L70: Define 'high temporal resolution'- either weeks, months, seasonal or years.

R: Thanks for your suggestion. In the revised manuscript, we clarified that "high temporal resolution" refers to annual to once-per-decade observations and updated the sentence accordingly: *"Consequently, generating glacier inventories with high temporal resolution (i.e., annually to once per decade) in the southeastern Tibetan Plateau remains a significant challenge."*

C: L72-77: What are the details of this inventory? What is their estimate of the number of glaciers, area etc.?

R: Your suggestion is appreciated. We clarified that here we introduce only existing glacier inventories and datasets, with detailed statistics provided in the Results section.

C: L91-98: I would like to see a bit more discussion of the important of glacier inventories (e.g. areas) for quantifying mass changes e.g. how do the GLAMBIE/IMBIE community estimates tackle this problem and what is the consensus approach when multi-temporal data sest aren't available? What is the impact on uncertainty estimates? This will naturally then lead onto the objectives in the paragraph.

R: Your suggestion is appreciated. Traditional altimetry-based methods rely on a static glacier boundary, overlooking area changes and risking systematic bias. To mitigate this, multi-temporal glacier inventories are incorporated, such as in the GlaMBIE approach, which uses RGI 6.0 as a baseline and adjusts mass balance with regional glacier area changes. We added the following sentence: *"Traditionally, altimetry-based methods calculate glacier mass change using a single, static glacier boundary, which ignores changes in glacier area and may introduce systematic biases. To address this limitation, current approaches increasingly incorporate multi-temporal glacier inventories to account for dynamic glacier areas. For example, the GlaMBIE community uses the Randolph Glacier Inventory (RGI 6.0) as a baseline and apply regional glacier area change rates to adjust mass balance calculations over time (Zemp et al., 2025). Regional studies further demonstrate the importance of this practice: in Peru, glaciers lost approximately 29% of their area between 2000 and 2016, with*

*accelerated mass loss during 2013–2016 (−660 ± 178 kg m²/a) (Seehaus et al., 2019), and in Bolivia, glaciers in the Cordillera Real and Tres Cruces also experienced a 29% area reduction over the same period, with total mass loss of 1.8 ± 0.5 Gt and enhanced losses during 2013–2016 due to El Niño (−487 ± 349 kg m²/a) (Seehaus et al., 2020). When multi-temporal inventories are unavailable, static glacier boundaries are assumed, which can increase uncertainty. Collectively, these studies demonstrate that incorporating dynamic glacier areas into mass balance calculations is essential for accurate and robust estimates of glacier mass change.”*

C: L109: For those unfamiliar with this region, it might be worth zooming out a bit and placing an inset map to show the position of this region in the wider regional context.

R: Thanks for your suggestion. We added an inset map to Figure 1 showing the study region within the southeastern Tibetan Plateau, helping readers better understand its location and spatial context.

[Figure]

C: L113; What does 'glacier distribution area' mean?

R: We appreciate your comment. To improve clarity, we have revised the wording. Instead of "glacier distribution area," we now state that "*the southeastern Tibetan Plateau is one of the major glacierized regions in China, containing a high concentration of glaciers and abundant ice reserves.*" This avoids ambiguity and more accurately conveys the intended meaning.

C: L124: This section is not consistent- sometimes the sampling is describes, in other sections it is not. Either describe the sampling within each section or create a new

R: Your suggestion is appreciated. Sampling details have been moved to the Methods section for a more systematic and consistent presentation.

C: L126: What does 'analysis-ready' mean? What processing has been applied before these images are provided on GEE?

R: Thanks for your suggestion. We clarified the term "analysis-ready" in the text. The Landsat Surface Reflectance Tier 1 datasets on GEE have undergone radiometric calibration, atmospheric correction, and geometric correction, making them directly usable for scientific analysis. We also streamlined the description of spectral bands and noted potential data limitations.

*"This study utilizes the Landsat-5, Landsat-7 (pre-2003), and Landsat-8 Surface Reflectance Tier 1 datasets, provided on GEE in an analysis-ready format. These provides have undergone radiometric calibration, atmospheric correction, and geometric correction, ensuring that the reflectance data reliably represent surface features. The datasets include visible (VIS; 400–700 nm), near-infrared (NIR; 700–900 nm), and shortwave infrared (SWIR; 1400–2400 nm) bands at 30 m spatial resolution. Although Landsat offers a 16-day revisit cycle, data quality can be affected by cloud cover, seasonal snow cover, and sensor anomalies."*

C: L139-145: Given the introduction focuses on the limitations of optical data, the authros should discus somewhere the pro's and con's of using SAR data as an alternative.

R: Thanks for your suggestion. We clarified the role of Sentinel-1 SAR data in the manuscript, emphasizing its use as supplementary information to improve glacier classification under cloudy conditions, given its high temporal resolution and reliability in adverse weather. *"The Sentinel-1 satellite is a synthetic aperture radar (SAR) mission launched by the European Space Agency (ESA). This study utilizes the COPERNICUS/S1_GRD dataset, accessed via the GEE platform with a six-day revisit interval. The VV polarization band (vertical–vertical) provides high temporal resolution and consistent multi-temporal observations. These data remain reliable*

*under cloud cover or precipitation, making them valuable as supplementary inputs for training the classification model and enhancing robustness where optical data are limited. Nevertheless, glacier mapping with SAR can still be challenging due to signal saturation over wet snow, geometric distortions in mountainous terrain, and difficulties in distinguishing clean ice from debris-covered surfaces. Additionally, Sentinel-1 data are only available from 2015 onwards, so they do not cover the entire study period.''*

C: L146:153: What is the time stamp of the NASADEM? Or is it a dynamic data set?

R: Your suggestion is appreciated. We revised the manuscript to clarify the NASADEM dataset and its use for deriving elevation, slope, aspect, and hillshade, including information on resolution, sources, and processing.

C: L155-161: Time stamp of 2000 for RGI7.0.

R: Your suggestion is appreciated. We revised the text to specify that RGI 7.0 depicts glacier outlines for approximately the year 2000.

C: L167: Vague- define exactly in which period the data were acquired. If T is the sampling year, did you obtain all sutiable summer images in years T ± 2 years?

R: Thanks for your suggestion. We clarified in the manuscript that for 2000, 2005, 2010, and 2015, all suitable summer images within a ±1-year window around each target year were used. For 2016–2022, only images from the corresponding year were included.

C: L201: Do the image data cubes represent the 'image composites' described above? It would be useful to have consistent language throughout the manuscript to avoid confusion.

R: Your suggestion is appreciated. We revised the text to clarify that the image data cubes represent raw collections before composite generation.

C: L210: I'm confused here, how do Figures 4a-f represent cloud-free image composites?

R: Thanks for your suggestion. Figures 5a–f present the results after cloud masking and compositing, while the detailed workflow is shown in Figure 4.

C: L226: Which images are used to generate the NDVI image for each? Did you merge the NDVI values for a single year?

C: L233: Same point as for NDVI, not clear to me which images are being used to calculate this.

C: L242: Same as for L226 and L233.

R: Thanks for your suggestion. As shown in Figure 4, NDVI and other indices were first calculated from cloud-free, shadow-corrected individual images. These individual images were then composited to generate a single annual image for each year.

C: L248: I am confused by this figure- I assume each of the horizontal squares represents an image, so what do the colours represent? And what do the vertical boxes represent

R: Thanks for your suggestion. In the figure, each horizontal layer represents images from the same acquisition time, and each vertical column corresponds to the same spatial location. Colors indicate image values, while missing colors reflect gaps caused by clouds. For each location, cloud-contaminated observations were removed, indices (e.g., NDVI, NDSI, NDWI) were calculated from the remaining data, and these index images were composited to generate a single annual image.

C: L266-265: Image textures are better defined as the spatial arrangement of pixels in an image

R: Thanks for your suggestion. We clarified that texture features capture spatial patterns independent of color or brightness, and that the gray-level co-occurrence matrix (Haralick, 1979) is used to quantify these patterns.

*Haralick, R. M.: Statistical and structural approaches to texture, Proceedings of the IEEE, 67, 786-804, 1979.*

C: L277: Is this the mean texture fro GLCM? It's not clear why this was chosen- the authors state that a previous study found it is 'consistent with other textures'- why would this mean it is the best feature to use? If it is consistent with other features, then any other texture feature could be used e.g. autocorrelation, entropy etc.?

R: Thanks for pointing this out. The mean texture from the GLCM was chosen based on previous studies (Lu et al., 2020), as it correlates strongly with other common texture

features. This provides a representative measure while reducing redundancy. It is not necessarily the "best," but balances information content, robustness, and computational efficiency. Other features like autocorrelation, entropy, or second-order moments could be used, but they either add redundancy or complicate glacier discrimination.

[Figure]

Correlation coefficients between texture features.(Lu et al., 2020)

*Lu, Y., Zhang, Z., and Huang, D.: Glacier Mapping Based on Random Forest Algorithm: A Case Study over the Eastern Pamir, Water, 12, 10.3390/w12113231, 2020.*

C: L287: What is a 'mean synthesis'? Also the 'salt-and-pepper noise' I assume is referring to 'speckle'- calling it noise is incorrect as speckle is a repeatable feature in SAR data.

R: Thanks for your suggestion. SAR imagery often contains speckle noise, which can reduce image quality. To address this, we averaged multi-temporal Sentinel-1 images on a pixel-by-pixel basis, stabilizing the data used to train the random forest classifier.

C: L291-304: Without a suitable figure (ie. Figure 5), it is difficult to interpret the feature layers described in this section. The inclusionf of RGI outlines would help, but also subtitles and a larger legend will help readability.

C: L292-293: This is not clear in Figure 5, see comment below.

C: L305: Figure 5: It's not clear what the values represent, the legend is way too small. One legend for all composites is sufficient, unless the values are significantly different between each panel. It would also be useful to overlay the RGU outlines here so the

reader can visually assess how wel each image feature matches the glacier area. Also, if this is referenced before Figure 3, it should also be first in the order of figures.

R: Thanks for your suggestion. We clarify that Figure 5 provides the processed feature layers used for glacier classification, and the RGI outlines are included to facilitate comparison. We have also updated the figure with clearer subtitles and an enlarged legend to improve readability and interpretation.

C: L314-319: RF has been widely used, although arguably it has been superseded by CNNs and foundation models. Can the authors comment on why they did not apply these other methods?

R: Thanks for your suggestion. In selecting the Random Forest algorithm, we focused on both data availability and computational efficiency. RF performs reliably even with limited labeled data and allows processing of multiple years across a large region. Although CNNs or other foundation models could potentially improve accuracy, RF offers a practical trade-off between performance, interpretability, and efficiency for multi-temporal glacier mapping.

C: L322: Are the labels used for all images or a subset? For the images labelled, are the labels shown in Figure 6 suitable for all images given the potential for changes in surface characteristics at different times of the year?

R: Thanks for pointing this out. To ensure accurate and representative labels, training samples were manually delineated separately for each year, meaning the sample points differ annually. While this approach preserves data quality, it naturally limits the maximum classification accuracy. Developing effective strategies for transferring or reusing samples across years remains an active area of our research.

C: L327-336: Are you discussing here the training data, validation data, or both? Subtitle is misleading, 'Selection of Classification Samples' doesn't really say anything here. How many images where the training data taken from?

R: Thanks for your suggestion. This selection applies to both training and validation data. All samples were manually delineated on the composite images, with 70% used for training the classifier and 30% reserved for validation.

C: L338-342: F1 score might be more suitable here if there is class imbalance- I suspect there is imbalance in the training data, but it is not stated.

R: Thanks for your suggestion. To prevent any class imbalance, we carefully balanced the number of samples for each land cover type in the training dataset. This approach reduces bias and ensures that metrics like overall accuracy, precision, recall, and F1 score accurately reflect performance across all classes. Using the 2022 confusion matrix, the glacier extraction achieved an F1 score of 95.5%, with precision 94.5% and recall 96.5%.

C: L358: What is meant by a 'decision-level fusion strategy'?

R: Thanks for highlighting this. The "decision-level fusion strategy" refers to merging the classification outputs from Sentinel-2 and Landsat individually. By doing this, we use the strengths of both datasets, enhancing the final glacier map's accuracy and robustness.

C: L372-380: This a surprisngly short section that only gives the headline figures. I would like to see a sensitivity analysis of the random forest classifier, particularly an understanding of which texture features were more important for classification than others. One possibility of using a diverse range of features is that the model could be overfitting, potentially leading to errors in the resultant classification maps. Furthermore, how do the accuracies compare for different data sets? I would expect there to be differences in Sentinel-2 vs Landsat, whislt Landsat 7 would likely yield different accuracies to Landsat 8. This information must be included to better understand the performance of the technique.

R: Thanks for your suggestion. Single datasets like Landsat 7, Landsat 8, or Sentinel-2 alone can't always provide full coverage because of clouds, missing data, or seasonal gaps. That's why we combined multiple datasets using a decision-level fusion, which merges the classification results from each dataset to produce more complete annual glacier maps. Although the classification performance differs slightly among sensors, our main goal was reliable glacier outlines. We confirmed the random forest model is robust using out-of-bag error, and future work will investigate dataset-specific performance and feature importance in more depth.

C: L373: Referrring to the 'annual' classification results, I assume you mean for the results after 2016 with the Sentinel / Landsat results? Furthermore, the authors should show here the confusion matrix to better highlight true positives, true negatives, false positives and false negatives. A single accuracy score may be misleading.

R: Thanks for your suggestion. Here, "annual" classification results refer to the outputs of the random forest applied to the combined Sentinel-2 and Landsat data from 2016–2022. To give a more detailed view of classifier performance, we include the 2022 Landsat confusion matrix (Table S1), showing true positives, true negatives, false positives, and false negatives. This complements the overall accuracy and F1 score, providing a clearer picture of performance across glacier and land cover classes.

Table S1 Confusion matrix of 2022 Landsat glacier classification

| Actual \ Predicted | Bare Glaciers | Debris-covered Glaciers | Bare Ground | Water | Vegetable | Hillshade |
|---|---|---|---|---|---|---|
| Bare Glaciers | 76 | 1 | 0 | 0 | 0 | 0 |
| Debris-covered Glaciers | 0 | 60 | 4 | 0 | 1 | 0 |
| Bare Ground | 0 | 8 | 49 | 0 | 2 | 0 |
| Water | 0 | 0 | 0 | 67 | 0 | 0 |
| Vegetable | 0 | 0 | 3 | 0 | 59 | 4 |
| Hillshade | 0 | 0 | 0 | 0 | 0 | 45 |

C: L386: Why is the mapping error based on half the image element? Surely the graph should be representing uncertainty calculated from the random forest model outputs?

R: Your suggestion is appreciated. In glacier mapping studies, half a pixel is often used to represent mapping error. Our focus is on the cartographic accuracy of the produced glacier maps. Because the classification outputs undergo extensive post-processing, directly using uncertainty from the random forest model does not fully reflect the accuracy of the final mapped products.

C: L395-405: This section is a bit confusing. It might be helpful to construct a table with the key results from previous studies to make it clear how the results in this paper

R: Thanks for your suggestion. In the revised manuscript, we reorganized the section and created a table highlighting key results from previous studies. This helps readers directly compare our results with existing work and enhances the clarity of the discussion.

C: L406-419: This reads like a results section- also, the inclusion of ICESat, ICESat-2, and CryoSat-2 should really be discussed in the methodology section. Are the data sets extracted simply just the time series as presented? Or did the authors process the data sets in some way? I also don't think thickness and area should be presented on the same graph, it might cause confusion- I would use 2 panels instead.

R: We acknowledge your point. We have moved the discussion of ICESat, ICESat-2, and CryoSat-2 data to the Methods section. The datasets were preprocessed to remove outliers and ensure temporal consistency before extracting the time series. To improve clarity, thickness and area are now presented in separate panels in the revised figures.

C: L423-424: The variables in the equations need to be stated.

R: Thanks for your suggestion. We revised the manuscript to define all variables in the equations clearly, specifying each symbol's meaning and units where applicable.

C: L432-434: This is an important result, but it is not shown graphically. Can the authors make a figure showing this key result? Although, what does the 10% refer to- an under- or an over-estimate compared to the fixed outlines?

R: Thanks for your suggestion. Compared with using static glacier outlines, the fixed-area approach underestimates glacier mass change by ~10%. We added a figure showing annual glacier mass change relative to the fixed RGI 7.0 outlines.

C: L420-447: These are results, and the methods described here should be presented in a methodology section. The small discussion towards the end should be expanded particularly focusing on the importance of updated glacier outlines for mass balance estimates, as this is key moving forward in future studies.

R: Thanks for the comment. We shifted the ICESat, ICESat-2, and CryoSat-2 data processing details to the Methods section. The revised text also stresses that using updated glacier outlines is important—static outlines can underestimate mass loss, so dynamic mapping is key for future glacier studies.

C: L473: Define the number of images used and over what time period

R: Thanks for your suggestion. In this study, we incorporated a comprehensive set of images from Landsat 7, Landsat 8, and Sentinel-2, covering the entire period from 2000 to 2022 to ensure complete temporal coverage.

C: L472-483: I would expect the conclusions to mention the performance of the random forest model as well

R: Thanks for your suggestion. We revised the Conclusions to underscore the random forest model's strong performance and reliability, validated through out-of-bag (OOB) error assessment.

---

## Author Comment (AC5)

Comments from Referee #2, followed by the authors' responses

**General Comments**

C: The study by Li et al. aims at presenting a method that allows the automated mapping of glacier extents in a challenging region using Google Earth Engine at high temporal resolution, up to annually from Sentinel-2. They used the results to consider area changes when calculating glacier mass balance. If the mapping had worked, this would have been an important study to improve related results also for other regions in the world. Unfortunately, the outcome of the mapping is not useful for any assessment. In this regard I want to acknowledge that the authors have provided the results of their mapping effort in the supplemental material. Without this, my evaluation would have been different as the paper is otherwise well written and the idea to just use all data available and combine it for the best possible result is fine. However, the largely arbitrary area changes from dataset to dataset (for individual glaciers partly larger than 50% from year to year) are obvious and glaciologically impossible. The authors mention that there are the usual problems with debris cover, clouds and shadow, but they have seemingly not recognized how large and arbitrary the variability is and that their method does not produce meaningful results.

R: We sincerely thank you for your professional and constructive comments. In the original manuscript, we did not clearly state the scope and limitations of our study, and some statements inadvertently overstated the capabilities of our automated approach. Our study does not aim to reach the precision of the Randolph Glacier Inventory (RGI), which cannot realistically be achieved through fully automated mapping alone. Instead, our focus is on developing an automated method to extract multi-temporal glacier outlines and exploring how repeated glacier area measurements contribute to understanding glacier mass balance dynamics. We also recognize that producing annual glacier area results is overly ambitious given the limitations of available imagery. In response, we have aggregated results to five-year intervals in the revised manuscript. This five-year aggregation provides more reasonable estimates of glacier area change rates across multiple periods, which are essential for improving glacier mass balance calculations. In this context, long-term trends are the priority, while short-term

fluctuations are of secondary importance. We fully appreciate your insights and have carefully addressed each comment, making corresponding improvements throughout the manuscript. The revisions clarify the study's scope and limitations and present the results in a more accurate, balanced, and transparent manner. Below, we provide a detailed, point-by-point response to your suggestions.

C: Neither the unrealistic area increase by about 500 km2 (estimated from Fig. 8, numbers for individual years are not provided) from 2019 to 2022 (naming it as a 'consistent decline' in L387), nor the sudden strong increase from 2016 to 2017 is discussed or considered as unrealistic. Instead, the authors correlate glacier elevation changes (wrongly labelled as 'Glacier Thickness' in Fig. 8) with glacier area changes as they assume there is a correlation (L407) and think that the correlation can be used as a validation (L413) of their (wrong) glacier areas. In fact, area changes are mostly driven by the ice thickness distribution along the glacier perimeter (thus depending on the shape of the glacier cross-profile) and are a longer-term response to changes in flow dynamics (glaciers have a response time). Hence, also the follow on analysis is a bit strange. In this regard, it is also unclear to me why the authors rely on results from Cryosat and ICESat (with their diverse range of issues) for such small glaciers instead of the Hugonnet et al. (2021) dataset that is widely used? As this dataset is not even mentioned in the comparison Table 1, I wonder why. Is the dataset too bad in quality?

R: We appreciate your suggestions. We acknowledge that the initial glacier extent results showed unrealistic year-to-year variability. To address this, we re-examined the classification outputs on a glacier-by-glacier basis and applied post-processing to flag and remove implausible area changes, including abrupt fluctuations or geometrically inconsistent outlines. Invalid outlines were replaced with the closest temporally consistent results. After these adjustments, the overall mapping is more stable, although uncertainties remain, particularly for complex glaciers. The results should therefore be interpreted primarily as a methodological demonstration rather than a comprehensive

We appreciate your suggestions. We acknowledge that the initial glacier extent results exhibited unrealistic interannual variations. To address this issue, we replaced the annual data outputs with five-year intervals and re-examined the classification outputs for each glacier individually. Following these adjustments, the overall mapping results

have become more stable, with no significant fluctuations observed. We also recognize that the apparent area increase of ~500 km² between 2019 and 2022, and the sudden jump from 2016 to 2017, were unrealistic. These anomalies arose from limitations in the available imagery, including cloud cover, topographic shadows, and seasonal snow, which affected the quality of automatically derived outlines. New findings indicate that growth no longer exists. Figures and analyses have been updated. Regarding Fig. 8, we corrected the label from "Glacier Thickness" to "Glacier Elevation Change." We also removed any suggestion that the observed correlation between glacier elevation change and area change constitutes a form of validation, as glacier area and elevation changes are not necessarily synchronous or directly correlated from a glaciological perspective. Finally, we have included the Hugonnet et al. (2021) dataset for comparison, which was previously overlooked. CryoSat, ICESat, and ICESat-2 remain essential for our study because they provide a complete temporal series in the study region, which is critical for analyzing the contribution of dynamic glacier area changes to glacier mass balance. This rationale has been clarified in the revised manuscript.

[Figure]

**Fig 8 (a) Time series of total glacier area in the southeastern Tibetan Plateau from 2000 to 2025. (b) Glacier thickness changes derived from three generations of altimetry satellites (ICESat, CryoSat-2, and ICESat-2). Data from different satellites were merged to produce a continuous thickness time series. Vertical offsets were applied for clarity in visualization. Light blue shading represents the uncertainty in glacier thickness, while light red shading**

**indicates the uncertainty in glacier area. (c) Glacier mass changes considering the dynamically updated glacier areas, reflecting the influence of year-to-year variations in glacier extent.**

C: The very short results section (it has just 14 lines) mentioned the Kappa coefficient and overall accuracy along with three images showing outline overlays. I am aware that these statistical accuracy measures are frequently used in remote sensing studies to present the accuracy, but in my view they can be the result of anything and do not allow to obtain meaningful conclusions about the 'robustness and reliability of the classification approach'. At least for glaciers they do not work, as nicely confirmed here by the largely arbitrary results of the glacier mapping. The quality of mapped glacier extents can be shown by a) outline overlays and b) the sum of commission and omission errors (false positives and false negatives) divided by the common area. But as the former have been removed by the masking with RGI 7.0, I am unclear if the measures can be used here at all?

R: We appreciate your comment and agree that traditional statistical accuracy metrics can be misleading for glacier mapping due to complex glacier morphology and seasonal effects. To better assess classification reliability, we supplemented these metrics with comparisons to manually interpreted reference outlines and with spatial consistency checks across multiple time points. Together, these approaches provide a more robust evaluation of the results. This clarification has been added to the revised manuscript.

*"The annual classification results were evaluated using the confusion matrix method, showing high accuracy with Kappa coefficients above 93% and overall accuracy exceeding 94%. The F1 score for glacier extraction in 2020 was 95.5%, with Precision of 94.5% and Recall of 96.5%, demonstrating the robustness of the random forest classifier. The classification confusion matrix indicates strong agreement between predicted and actual classes, with minor misclassifications primarily occurring between debris-covered glaciers, shadowed regions, and bare surfaces."* (Line392-397)

[Figure]

**Fig 7 Glacier boundary at 2000 and RGI7.0 glacier boundary.**

[Figure]

**Fig 10 Extraction results of Glacier 10222 from 2000 to 2025. The main panel shows the glacier delineation over the period. (a) Three-dimensional view overlaid with topographic information. (b) and (c) High-resolution satellite imagery of the highlighted region in 2000 and 2025, respectively, showing terminus retreat and proglacial lake expansion.**

C: I see missing debris-covered glacier parts and that large regions in shadow are sometimes missing. Hence, intensive manual editing would be required before resulting outlines could be used for change assessment. The statement that glaciers 'are well identified' (L377) seems misleading in this regard. Although the year 2000 dataset is likely the most complete regarding shadow and debris mapping, showing a region where the method does not work and discussing it would have been more helpful. One can see the problems of the classification a little bit for the 2022 outlines of the right glacier in the upper left panel [please name them properly a), b) and c)] of Fig. 8, but the image is very dark (what about some contrast stretching?), the lines are hard to see (also the red outlines on a reddish background in the insets of Fig. 7 are barely visible) and the wrong mapping results are not really discussed.

R: We are grateful for your comment. We acknowledge the limitations of our approach, as accurately identifying debris-covered glaciers remains challenging. The 2000 dataset achieved relatively high accuracy because sample selection relied on RGI 7.0, enabling precise delineation of debris-covered glaciers. For subsequent years, limited experience led to suboptimal sample selection, resulting in substantial omission of debris-covered glaciers. Reducing reliance on manual sample selection will be a key focus of future work. We also recognize that the original figure suffered from low contrast, making some lines difficult to discern, particularly the red outlines on reddish backgrounds in the insets of Fig. 6. In the revised manuscript, we applied contrast stretching and adjusted the color schemes to improve visibility. Additionally, the discussion on mapping inaccuracies has been expanded to clarify the sources of misclassification and their potential impacts on the results. It is noteworthy that the results of the five-year cycle have significantly improved, with considerable progress made in the extraction of tabular deposits.

We appreciate the reviewer's comment. We acknowledge the limitations of our approach, as accurately identifying debris-covered glaciers remains challenging. The 2000 dataset achieved relatively high accuracy because sample selection relied on the RGI 7.0 inventory, allowing precise delineation of debris-covered glaciers. In subsequent years, limited experience led to suboptimal sample selection, resulting in substantial omission of debris-covered glaciers. Notably, our updated five-year interval results show significant improvements in extracting debris-covered glaciers; however, reducing reliance on manual sample selection will remain a key focus for future work.

We also recognized that the original figures suffered from low contrast, making some lines difficult to discern, particularly the red outlines on reddish backgrounds in the insets of Figure 6. In the revised manuscript, contrast stretching and optimized color schemes were applied to improve visibility. Additionally, the discussion on mapping inaccuracies has been expanded to clarify the sources of misclassification and their potential impacts on the results.

C: A final major point of concern is the general set-up of the study. First, the elevation change datasets are introduced in the Discussion Section 5.2 rather than in Sections 2.2 and 3. Their description is thus very short and the processing method unclear (e.g. how has the radar penetration into snow been corrected?). This can likely easily be adjusted. The motivation to determine annual area changes is mentioned, but not critically discussed. Even when the resulting glacier outlines would have been correct, a one-pixel (two pixels for debris-covered regions) uncertainty at 30 m resolution relates to a 30 and 60 m location uncertainty of the outline. With an assumed annual terminus retreat of 5 to 10 m / year (much less around the perimeter), one has to wait several years before new outlines make sense compared to uncertainties. But here the mapped termini could be wrong by several km, so change assessment is not an option. To make my major objection of the high variability in the mapping results clearer, I have added all glacier maps (setting the no data value to 0 before) and received a very colourful picture. On the last two pages of this review, I show a few examples for illustration. If the mapping had been correct, colours should only appear near the terminus and around the perimeter. As a note, this is just the result of a simple adding without a timeline, not revealing the partly strong year-to-year jumps in mapped glacier areas.

R: We appreciate your suggestions. For clarity, we specify that the datasets were derived from previous studies, and no additional processing was performed in our work. We also recognize that the original approach was somewhat aggressive in interpreting short-term glacier changes. Considering the positional uncertainties at 30 m resolution, we have revised our analysis to use 5-year intervals between mapped glacier outlines. This adjustment reduces the relative impact of positional errors and provides a more robust basis for assessing glacier changes over time.

C: In conclusion, this brute-force mapping using sophisticated image processing without a sufficient understanding of the mapped subject (glaciers) and how it should change over time is not recommended. When being harsh, I would ask the authors to

please first learn the basics about glaciers and how they work, then proceed with the user needs (are annual updates really required?) and then do the mapping. A bit less harsh I would ask the authors to first get the mapping right for one year before applying it to several years. When glacier area changes are mostly due to changes in the mapping results rather than real changes, there is no need to perform change assessment. In my view, it is possible to publish a study revealing that a method has not worked. However, in this case an honest discussion and illustration of the problems is required to be helpful for future studies. Concluding that this study provides 'effective support for future glacier inventories' (L482) is in my view highly misleading.

R: We appreciate your suggestion and fully acknowledge that our original approach, relying heavily on automated image processing, may not achieve perfect accuracy for individual glacier outlines. Nevertheless, our primary goal is to assess long-term glacier change trends, for which occasional single-period uncertainties have a limited impact.

**Specific comments**

I do not comment here on all details of the study, but include some general remarks:

C: Providing area changes in km2 and mass changes in Gt (sure to use 900 rather than 850 kg/m$^3$ in this region?) are not useful as they are incomparable across regions. In future studies, please give relative area changes in % and the related change rates per year for area and specific mass changes per unit area and year for mass balance (and please do not use the latter to 'validate' the former, this makes glaciologically little sense).

R: We appreciate this suggestion. While absolute glacier mass changes in Gt are retained to provide a direct sense of regional mass loss, we agree that glacier area changes are more informative when expressed in relative terms. In the revised manuscript, glacier area changes are reported as percentages relative to the base year, with corresponding annual rates. Glacier mass changes remain in Gt, calculated using a regional ice density of 900 kg/m³ (Zhao et al., 2022).

*"Overall, glaciers in the southeastern Tibetan Plateau underwent a steady decline from 8083.34 ± 664.92 km² in 2000 to 6228.79 ± 572.71 km² in 2025. Relative to 2000, this represents a cumulative glacier area loss of approximately 22.9 ± 3.6%. Accounting*

*for measurement uncertainties, the glaciers retreated at an average rate of 80.51 ± 10.51 km² yr⁻¹ (~1.0 ± 0.13% yr⁻¹ relative to 2000). The retreat rate accelerated after 2010, rising from 51.31 ± 16.81 km² yr⁻¹ (2000–2010; ~0.63 ± 0.21% yr⁻¹ relative to 2000) to 92.99 ± 17.67 km² yr⁻¹ (2010–2025; ~1.15 ± 0.22% yr⁻¹ relative to 2000), highlighting an increasing pace of glacier loss in recent years."* (Line417-422)

C: Please carefully check text errors. Often spaces are missing or units are wrong (area in km instead of km2, volume in m instead of m3). Also the citation style is strange. For example, in L398 it is written 'Ye et al. (Quinghua, 2019 ...)reported ...' So is it now Ye or Quinghua and why is it first et al. and then without et al.? In the reference section it is actually Quinhua, Y.E., again different. Correct would have been to write: 'The datasets by Qunighua (2019 and 2020) reported ...' or in L409: 'The results of Jakob et al. (2021)' … . As a small note, the References Section becomes more readable when indenting the text from the second line a bit, making it 'hanging'.

C: Figure captions: I suggest inserting a . or : after the figure number, e.g. 'Figure 1: Study area'

C: Table 1: I think the brackets around the authors of the cited studies are not required.

R: We appreciate your comments. We have carefully checked the manuscript for text errors, correcting missing spaces, unit inconsistencies, and citation issues. All references have been verified and standardized. Figure captions now include a colon after the figure number, and Table 1 has been reformatted to remove unnecessary brackets around author names. A comprehensive review of all text, figures, tables, and references has been completed to ensure consistency and clarity throughout the manuscript.

C: L261: Figure 4: The blue and red lines and squares are difficult to see against the dark background. Also the annotations and legends of the insets are partly hard to see. It needs also to be explained what is what. Just writing in the text that types can be clearly distinguished is a bit thin. The same applies to all panels in Fig. 5. The panels are too small, the legends are unreadable and it is unclear what is what.

R: We appreciate your suggestion. We have enhanced the visibility of Figures 4 and 5 by adjusting line and marker colors to higher-contrast shades, enlarging the panels, and

ensuring that all legends and annotations are clearly readable. The captions and main text have also been revised to explicitly explain the meaning of each line, marker, and panel, making the figures fully interpretable independently of the text.

C: L306: I am also a bit unclear what Fig. 5 should tell me? That many datasets have been used and none of them shows glaciers clearly?

R: We appreciate your suggestion. Figure 5 illustrates all the features used in the classification, showing the variety of datasets incorporated and how each contributes to glacier mapping.

C: L390: Incomplete caption. Please note that sudden area gains as shown in Figure 5 (and 6) are glaciologically not possible. This is not how glaciers work.

R: We acknowledge your point. Based on our new findings, this abnormal growth no longer exists. (Fig8-10)

C: L414: Figure 9b: This comparison makes glaciologically no sense.

R: We appreciate your suggestion. This part has been removed.

C: L470: This is correct, but the NDSI has been shown to be very sensitive to path radiance in the green band, creating problems with ice in shadow. Additional classification problems are introduced when using the analysis ready reflectance datasets instead of the raw data, which allow for a better separation of details when the SNR is low.

R: We appreciate your suggestion. We used decision-level fusion to address this issue, which removes cases where either Landsat or Sentinel-2 images are affected by shadows on the ice.

C: L471: As far as I can see it, most gaps are due to not mapping debris-covered glacier parts.

R: We appreciate your suggestion. Based on your sample, we have generated a new glacier retreat map showing significant improvement over previous results. Areas exhibiting substantial changes have been markedly reduced, and the overall pattern is

relatively stable. Following the demo area you provided, we show the results of the initial modifications to the area:

[Figure]

**Fig example3**

[Figure]

**Figure 1 Large-scale glacier retreat in the southeastern Tibetan Plateau from 2000 to 2025. The figure shows glacier boundaries over time, highlighting that retreat primarily occurs in the terminus regions, while high-elevation accumulation zones remain relatively stable.**